# A foundation model for generalizable disease detection from retinal images

Yukun Zhou[1,2,3 ✉], Mark A. Chia[2,4], Siegfried K. Wagner[2,4], Murat S. Ayhan[1,2,4], Dominic J. Williamson[1,2,4], Robbert R. Struyven[1,2,4], Timing Liu[2], Moucheng Xu[1,3], Mateo G. Lozano[2,5], Peter Woodward-Court[1,2,6], Yuka Kihara[7,8], UK Biobank Eye & Vision Consortium*, Andre Altmann[1,3], Aaron Y. Lee[7,8], Eric J. Topol[9], Alastair K. Denniston[10,11], Daniel C. Alexander[1,12] & Pearse A. Keane[2,4 ✉]

Medical artificial intelligence (AI) offers great potential for recognizing signs of health conditions in retinal images and expediting the diagnosis of eye diseases and systemic disorders[1]. However, the development of AI models requires substantial annotation and models are usually task-specific with limited generalizability to different clinical applications[2]. Here, we present RETFound, a foundation model for retinal images that learns generalizable representations from unlabelled retinal images and provides a basis for label-efficient model adaptation in several applications. Specifically, RETFound is trained on 1.6 million unlabelled retinal images by means of self-supervised learning and then adapted to disease detection tasks with explicit labels. We show that adapted RETFound consistently outperforms several comparison models in the diagnosis and prognosis of sight-threatening eye diseases, as well as incident prediction of complex systemic disorders such as heart failure and myocardial infarction with fewer labelled data. RETFound provides a generalizable solution to improve model performance and alleviate the annotation workload of experts to enable broad clinical AI applications from retinal imaging.

Medical artificial intelligence (AI) has achieved significant progress in recent years with the notable evolution of deep learning techniques[1,3,4]. For instance, deep neural networks have matched or surpassed the accuracy of clinical experts in various applications[5], such as referral recommendations for sight-threatening retinal diseases[6] and pathology detection in chest X-ray images[7]. These models are typically developed using large volumes of high-quality labels, which requires expert assessment and laborious workload[1,2]. However, the scarcity of experts with domain knowledge cannot meet such an exhaustive requirement, leaving vast amounts of medical data unlabelled and unexploited.

Self-supervised learning (SSL) aims to alleviate data inefficiency by deriving supervisory signals directly from data, instead of resorting to expert knowledge by means of labels[8–11]. SSL trains models to perform 'pretext tasks' for which labels are not required or can be generated automatically. This process leverages formidable amounts of unlabelled data to learn general-purpose feature representations that adapt easily to more specific tasks. Following this pretraining phase, models are fine-tuned to specific downstream tasks, such as classification or segmentation. The SSL model has outperformed supervised learning-based transfer learning (for example, pretraining the models with ImageNet[12] and categorical labels) in various computer vision tasks, even when the SSL models are fine-tuned with smaller amounts of data[13,14]. Besides this label efficiency, SSL-based models perform better than supervised models when tested on new data from different domains[15,16]. The combined qualities of strong generalization capacity of representations, and high performance achieved by fine-tuned models in many downstream tasks, indicate the great potential of SSL in medical AI in which data are abundant and healthcare tasks are diverse but labels are scarce[1,8].

Colour fundus photography (CFP) and optical coherence tomography (OCT) are the most common imaging modalities in ophthalmology and such retinal images accumulate quickly in routine clinical practice. In addition to illustrating clinical features associated with ocular diseases, these images also provide valuable insights into systemic diseases, a field that has recently been termed 'oculomics'[17,18]. For example, the optic nerve and inner retinal layers provide a non-invasive view of central nervous system tissue[19–21], and thus a window into neurodegeneration. Similarly, retinal vascular geometry provides insights into other vascular organ systems[22–25], such as the heart and kidneys. Although several studies have shown that SSL can increase performance for individual ocular disease detection tasks, such as the diagnosis of diabetic macular oedema[26], age-related macular degeneration (AMD)[27] and referable diabetic retinopathy[28–30], there has been limited work demonstrating the ability of a single SSL pretrained model to generalize to a diverse range of complex tasks. Progress has probably been hampered by the challenges involved with curating a large repository of retinal

[1]Centre for Medical Image Computing, University College London, London, UK. [2]NIHR Biomedical Research Centre at Moorfields Eye Hospital NHS Foundation Trust, London, UK. [3]Department of Medical Physics and Biomedical Engineering, University College London, London, UK. [4]Institute of Ophthalmology, University College London, London, UK. [5]Department of Computer Science, University of Coruña, A Coruña, Spain. [6]Institute of Health Informatics, University College London, London, UK. [7]Department of Ophthalmology, University of Washington, Seattle, WA, USA. [8]Roger and Angie Karalis Johnson Retina Center, University of Washington, Seattle, WA, USA. [9]Department of Molecular Medicine, Scripps Research, La Jolla, CA, USA. [10]Academic Unit of Ophthalmology, University of Birmingham, Birmingham, UK. [11]University Hospitals Birmingham NHS Foundation Trust, Birmingham, UK. [12]Department of Computer Science, University College London, London, UK. *A list of authors and their affiliations appears at the end of the paper. ✉e-mail: yukun.zhou.19@ucl.ac.uk; p.keane@ucl.ac.uk

images with extensive linkage to several relevant disease outcomes. Moreover, the capabilities of different SSL approaches (contrastive SSL versus generative SSL) and the interpretability of SSL models in retinal imaging, remain relatively under-explored. Developing an understanding of the specific features that SSL models learn during training is an important step for safe and reliable translation to clinical practice.

In this work, we present a new SSL-based foundation model for retinal images (RETFound) and systematically evaluate its performance and generalizability in adapting to many disease detection tasks. A foundation model is defined as a large AI model trained on a vast quantity of unlabelled data at scale resulting in a model that can be adapted to a wide range of downstream tasks[31,32]. Here we construct RETFound from large-scale unlabelled retinal images by means of SSL and use it to promote the detection of many diseases. Specifically, we develop two separate RETFound models, one using CFP and the other using OCT, by means of an advanced SSL technique (masked autoencoder[15]) successively on natural images (ImageNet-1k) followed by retinal images from the Moorfields diabetic image dataset (MEH-MIDAS) and public data (totalling 904,170 CFPs and 736,442 OCTs). We adapt RETFound to a series of challenging detection and prediction tasks by fine-tuning RETFound with specific task labels, and then validate its performance. We consider first the diagnostic classification of ocular diseases, including diabetic retinopathy and glaucoma; second, ocular disease prognosis, specifically conversion of contralateral ('fellow') eyes to neovascular ('wet') AMD in a 1-year time period and, finally, oculomic challenges, specifically the 3-year prediction of cardiovascular diseases (ischaemic stroke, myocardial infarction and heart failure) and a neurodegenerative disease (Parkinson's disease). RETFound achieves consistently superior performance and label efficiency in adapting to these tasks, compared to state-of-the-art competing models, including that pretrained on ImageNet-21k with traditional transfer learning. We also probe the interpretation of disease detection performance of RETFound with qualitative results and variable-controlling experiments, showing that salient image regions reflect established knowledge from ocular and oculomic literature. Finally, we make RETFound publicly available so others can use it as the basis for their own downstream tasks, facilitating diverse ocular and oculomic research.

Figure 1 gives an overview of the construction and application of RETFound. For construction of RETFound, we curated 904,170 CFP in which 90.2% of images came from MEH-MIDAS and 9.8% from Kaggle EyePACS[33], and 736,442 OCT in which 85.2% of them came from MEH-MIDAS and 14.8% from ref. 34. MEH-MIDAS is a retrospective dataset that includes the complete ocular imaging records of 37,401 patients with diabetes who were seen at Moorfields Eye Hospital between January 2000 and March 2022. After self-supervised pretraining on these retinal images, we evaluated the performance and generalizability of RETFound in adapting to diverse ocular and oculomic tasks. We selected publicly available datasets for the tasks of ocular disease diagnosis. Details are listed in Supplementary Table 1. For the tasks of ocular disease prognosis and systemic disease prediction, we used a cohort from the Moorfields AlzEye study (MEH-AlzEye) that links ophthalmic data of 353,157 patients, who attended Moorfields Eye Hospital between 2008 and 2018, with systemic disease data from hospital admissions across the whole of England[35]. We also used UK Biobank[36] for external evaluation in predicting systemic diseases. The validation datasets used for ocular disease diagnosis are sourced from several countries, whereas systemic disease prediction was solely validated on UK datasets due to limited availability of this type of longitudinal data. Our assessment of generalizability for systemic disease prediction was therefore based on many tasks and datasets, but did not extend to vastly different geographical settings. Details of the clinical datasets are listed in Supplementary Table 2 (data selection is introduced in the Methods section).

We compared the performance and label efficiency of RETFound against three pretrained comparison models: SL-ImageNet, SSL-ImageNet and SSL-Retinal. All models use differing pretraining strategies but have the same model architecture as well as fine-tuning processes for downstream tasks (architecture details are introduced in the Methods section). SL-ImageNet uses traditional transfer learning, that is, pretraining the model by means of supervised learning on ImageNet-21k (about 14 million natural images with categorical labels); SSL-ImageNet pretrains the model by means of SSL on ImageNet-1k (about 1.4 million natural images) and SSL-Retinal pretrains the model using SSL on retinal images from scratch. RETFound uses the weights of SSL-ImageNet as a baseline before extending to retinal images (equivalent to pretraining the model by means of SSL successively on natural images followed by retinal images). The pretraining schematics are shown in Extended Data Fig. 1. Furthermore, we explored the performance of using different SSL strategies, that is, generative SSL versus contrastive SSL approaches, by substituting the primary SSL technique (that is, masked autoencoder) for SimCLR[16], SwAV[37], DINO[38] and MoCo-v3 (ref. 14) within the RETFound framework, respectively. We reported internal and external evaluation results for these models. The models were adapted to each task with labelled training data, and evaluated on both held-out internal test sets, as well as external datasets completely distinct from the training data (details are listed in the Methods section). Model performance was reported using the area under the receiver operating curve (AUROC) and area under the precision-recall curve (AUPR). We calculated $P$ values with the two-sided $t$-test between RETFound and the most competitive comparison model for each task to check for significance.

## Ocular disease diagnosis

We included eight publicly available datasets to verify the model's performance on several ocular diseases and imaging conditions (Fig. 2). RETFound generally achieved the best performance in most datasets and SL-ImageNet ranked second, as shown in Fig. 2a. For instance, on diabetic retinopathy classification, RETFound achieved AUROC of 0.943 (95% confidence interval (CI) 0.941, 0.944), 0.822 (95% CI 0.815, 0.829) and 0.884 (95% CI 0.88, 0.887), respectively, on Kaggle APTOS-2019, IDRID[39] and MESSIDOR-2 (refs. 40,41) datasets, significantly outperforming SL-ImageNet (all $P < 0.001$). The superior performance can also be observed for glaucoma and the classification of many diseases. The AUPR results of RETFound were also significantly higher than the compared groups (Extended Data Fig. 2a). For external evaluation, we evaluated the performance of RETFound on diabetic retinopathy datasets (Kaggle APTOS-2019, IDRID and MESSIDOR-2), which were both labelled on the basis of the five-stage International Clinical Diabetic Retinopathy Severity scale. We conducted cross evaluation among the three datasets, that is, fine-tuned models on one dataset and evaluated them on the others. RETFound achieved the best performance in all cross evaluations, as shown in Fig. 2b. For instance, when fine-tuned on Kaggle APTOS-2019, RETFound achieved AUROC of 0.822 (95% CI 0.815, 0.829) and 0.738 (95% CI 0.729, 0.747), respectively, on IDRID and MESSIDOR-2 datasets, statistically significantly higher than SL-ImageNet ($P < 0.001$) on IDRID and SSL-ImageNet ($P < 0.001$) on MESSIDOR-2. The AUPR results of all groups were low but RETFound achieved significantly higher performance (Extended Data Fig. 2b). All quantitative results are listed in Supplementary Table 3.

## Ocular disease prognosis

For 1-year prognosis of fellow eye converting to wet-AMD, we evaluated the internal performance on data from AlzEye (Fig. 2c). With CFP as the input modality, RETFound showed the best performance with an AUROC of 0.862 (95% CI 0.86, 0.865), significantly outperforming the comparison groups ($P < 0.001$). The runner-up SL-ImageNet achieved an AUROC of 0.83 (95% CI 0.825, 0.836). With OCT, RETFound scored the highest AUROC of 0.799 (95% CI 0.796, 0.802), showing a statistically significantly higher AUROC ($P < 0.001$) than SSL-Retinal.

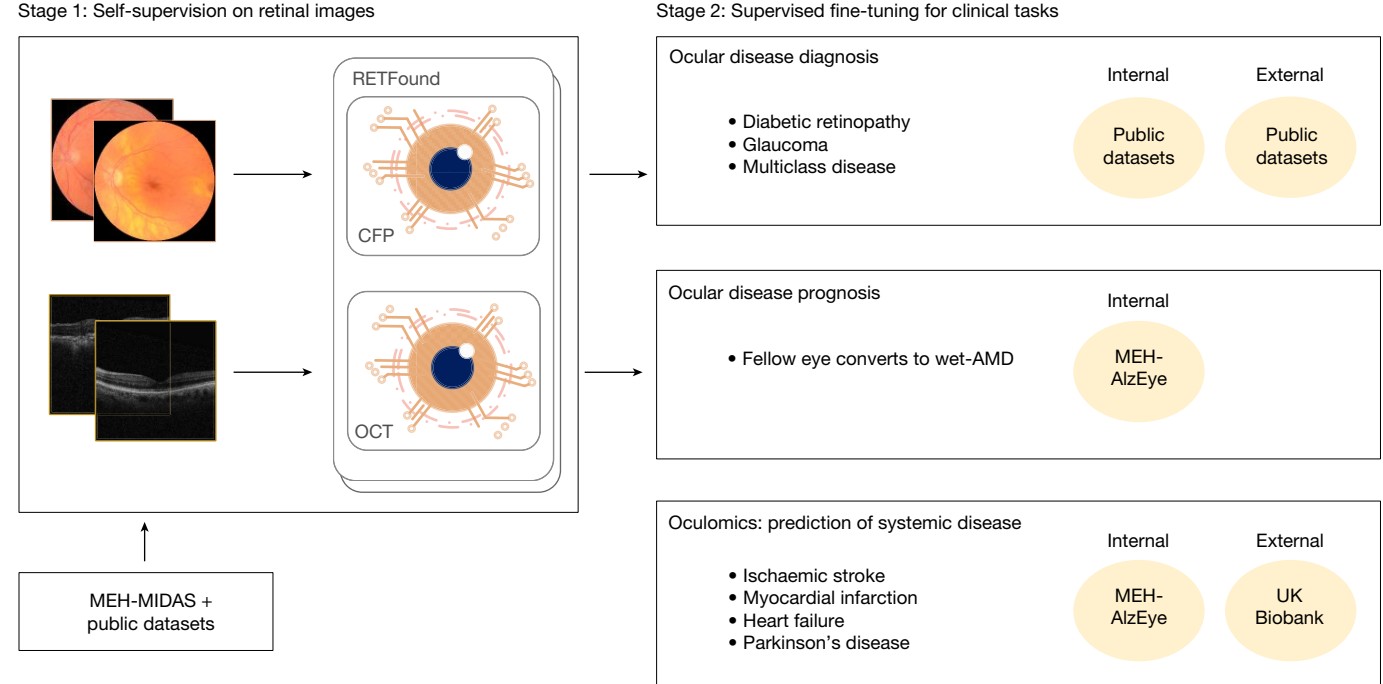

**Fig. 1 | Schematic of development and evaluation of the foundation models (RETFound).** Stage one constructs RETFound by means of SSL, using CFP and OCT from MEH-MIDAS and public datasets. Stage two adapts RETFound to downstream tasks by means of supervised learning for internal and external evaluation.

The AUPR results of RETFound are highest with CFP and comparable to SSL-Retinal with OCT (Extended Data Fig. 2c).

## Systemic diseases prediction

We organized four oculomic tasks to evaluate the model performance in predicting the incidence of systemic diseases with retinal images (Fig. 3). Although the overall performance was limited in these challenging tasks, RETFound has shown significant improvement in internal evaluation for both CFP and OCT, as shown in Fig. 3a. For the prediction of myocardial infarction with CFP, RETFound achieved AUROC of 0.737 (95% CI 0.731, 0.743). SSL-Retinal scored the second-best performance but was significantly worse than RETFound ($P < 0.001$). The confusion matrix (Extended Data Table 1) shows that RETFound achieved the highest sensitivity of 0.7 and specificity of 0.67. Likewise, RETFound also ranked first for prediction of heart failure, ischaemic stroke and Parkinson's disease with AUROCs of 0.794 (95% CI 0.792, 0.797), 0.754 (95% CI 0.752, 0.756) and 0.669 (0.65, 0.688), respectively. RETFound also performed significantly better than the other models when using OCT as the input modality. It achieved significantly higher AUPR results in all tasks (Extended Data Fig. 3a). External evaluation on the UK Biobank (Fig. 3b) showed that RETFound and SSL-Retinal performed similarly in prediction of ischaemic stroke. For tasks of myocardial infarction, heart failure and Parkinson's disease, RETFound achieved the best performance both with CFP and OCT. RETFound also showed significantly higher AUPR in most tasks when it was externally evaluated on UK Biobank (Extended Data Fig. 3b).

## Label efficiency for disease detection

Label efficiency refers to the amount of training data and labels required to achieve a target performance level for a given downstream task, which indicates the annotation workload for medical experts. RETFound showed superior label efficiency across various tasks (Fig. 4). For heart failure prediction, RETFound outperformed the other pretraining strategies using only 10% of labelled training data,

demonstrating the potential of this approach in alleviating data shortages. RETFound similarly showed superior label efficiency for diabetic retinopathy classification and myocardial infarction prediction. Furthermore, RETFound showed consistently high adaptation efficiency (Extended Data Fig. 4), suggesting that RETFound required less time in adapting to downstream tasks. For example, RETFound can potentially save about 80% of the training time required to achieve convergence for the task of predicting myocardial infarction, leading to significant reductions in computational costs (for example, credits on Google Cloud Platform) when appropriate mechanisms such as early stopping are used.

## SSL strategies for RETFound

We explored the performance of different SSL strategies, that is, generative SSL (for example, masked autoencoder) and contrastive SSL (for example, SimCLR, SwAV, DINO and MoCo-v3), in the RETFound framework. As shown in Fig. 5, RETFound with different contrastive SSL strategies showed decent performance in downstream tasks. For instance, RETFound with DINO achieved AUROC of 0.866 (95% CI 0.864, 0.869) and 0.728 (95% CI 0.725, 0.731), respectively, on wet-AMD prognosis (Extended Data Fig. 5) and ischaemic stroke prediction (Fig. 5), outperforming the baseline SL-ImageNet (Supplementary Tables 3 and 4). This demonstrates the effectiveness of RETFound framework with diverse SSL strategies. Among these SSL strategies, the masked autoencoder (primary SSL strategy for RETFound) performed significantly better than the contrastive learning approaches in most disease detection tasks (Fig. 5 and Extended Data Fig. 5). All quantitative results are listed in Supplementary Table 4.

## Model interpretation

To gain insights into the inner-workings of RETFound leading to its superior performance and label efficiency in downstream tasks, we performed qualitative analyses of the pretext task used for self-supervised pretraining and task-specific decisions of RETFound

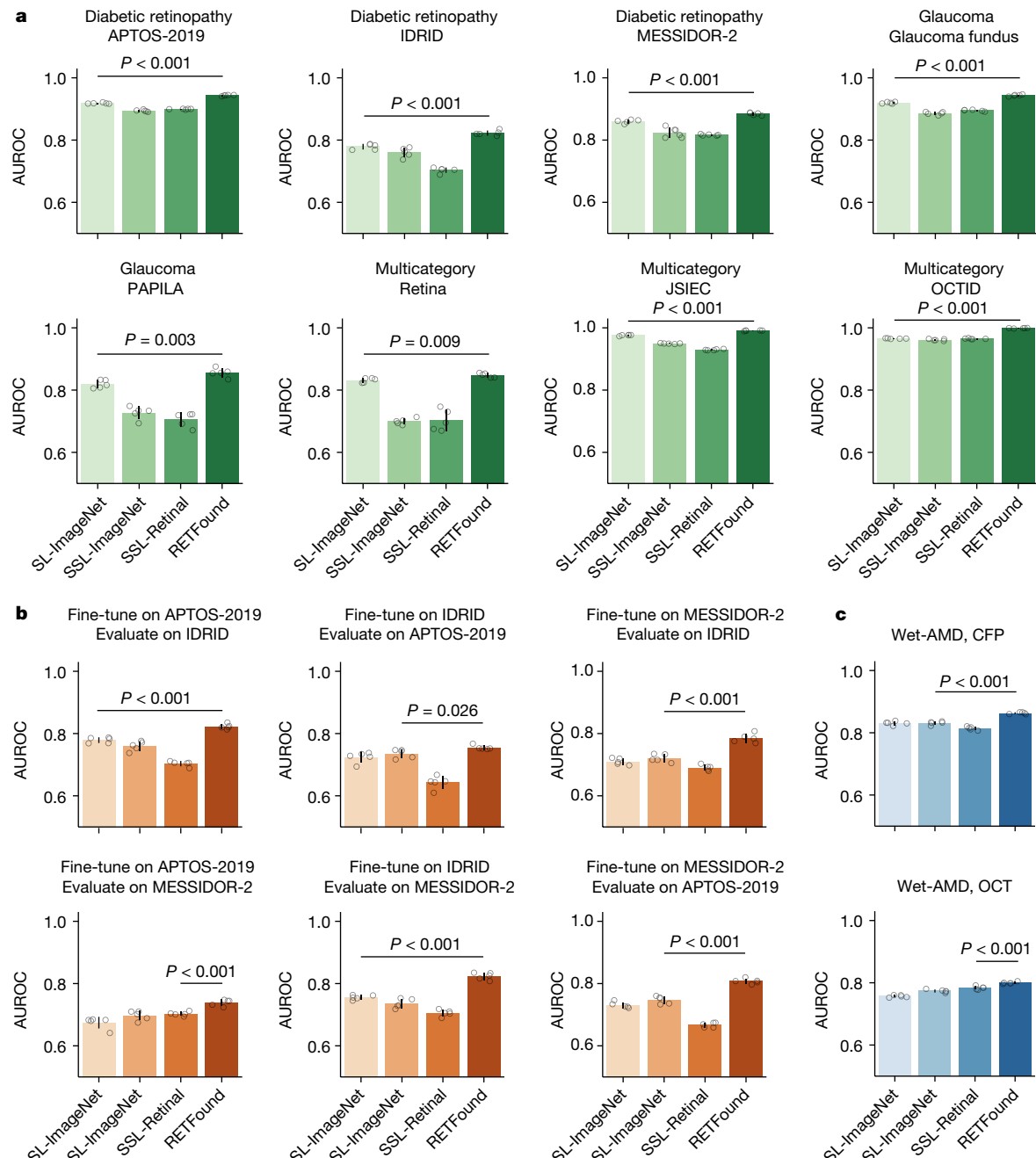

**Fig. 2 | Performance on ocular disease diagnostic classification. a**, Internal evaluation. Models are adapted to each dataset by fine-tuning and internally evaluated on hold-out test data in the tasks of diagnosing ocular diseases, such as diabetic retinopathy and glaucoma. The disease category and dataset characteristics are listed in Supplementary Table 1. **b**, External evaluation. Models are fine-tuned on one diabetic retinopathy dataset and externally evaluated on the others. **c**, Performance on ocular disease prognosis. The models are fine-tuned to predict the conversion of fellow eye to wet-AMD in

1 year and evaluated internally. RETFound performs best in all tasks. For each task, we trained the model with five different random seeds, determining the shuffling of training data, and evaluated the models on the test set to get five replicas. We derived the statistics with the five replicas. The error bars show 95% CI and the bar centre represents the mean value of the AUROC. We compare the performance of RETFound with the most competitive comparison model to check whether statistically significant differences exist. P value is calculated with the two-sided t-test and listed in the figure.

(Extended Data Fig. 6). The pretext task of RETFound allows models to learn retina-specific context, including anatomical structures and disease lesions. As shown in Extended Data Fig. 6a, RETFound was able to reconstruct major anatomical structures, including the optic nerve and large vessels on CFP, and the nerve fibre layer and retinal pigment epithelium on OCT, despite 75% of the retinal image being masked. This demonstrates that RETFound has learned to identify and infer the representation of disease-related areas by means of SSL, which

contributes to performance and label efficiency in downstream tasks. On top of the reconstruction-based interpretation, we further used an advanced explanation tool (RELPROP[42]) to visualize the salient regions of images conducive to classifications made by fine-tuned models in downstream tasks (Extended Data Fig. 6b). For ocular disease diagnosis, well-defined pathologies were identified and used for classification, such as hard exudates and haemorrhage for diabetic retinopathy and parapapillary atrophy for glaucoma. For oculomic tasks, we observed

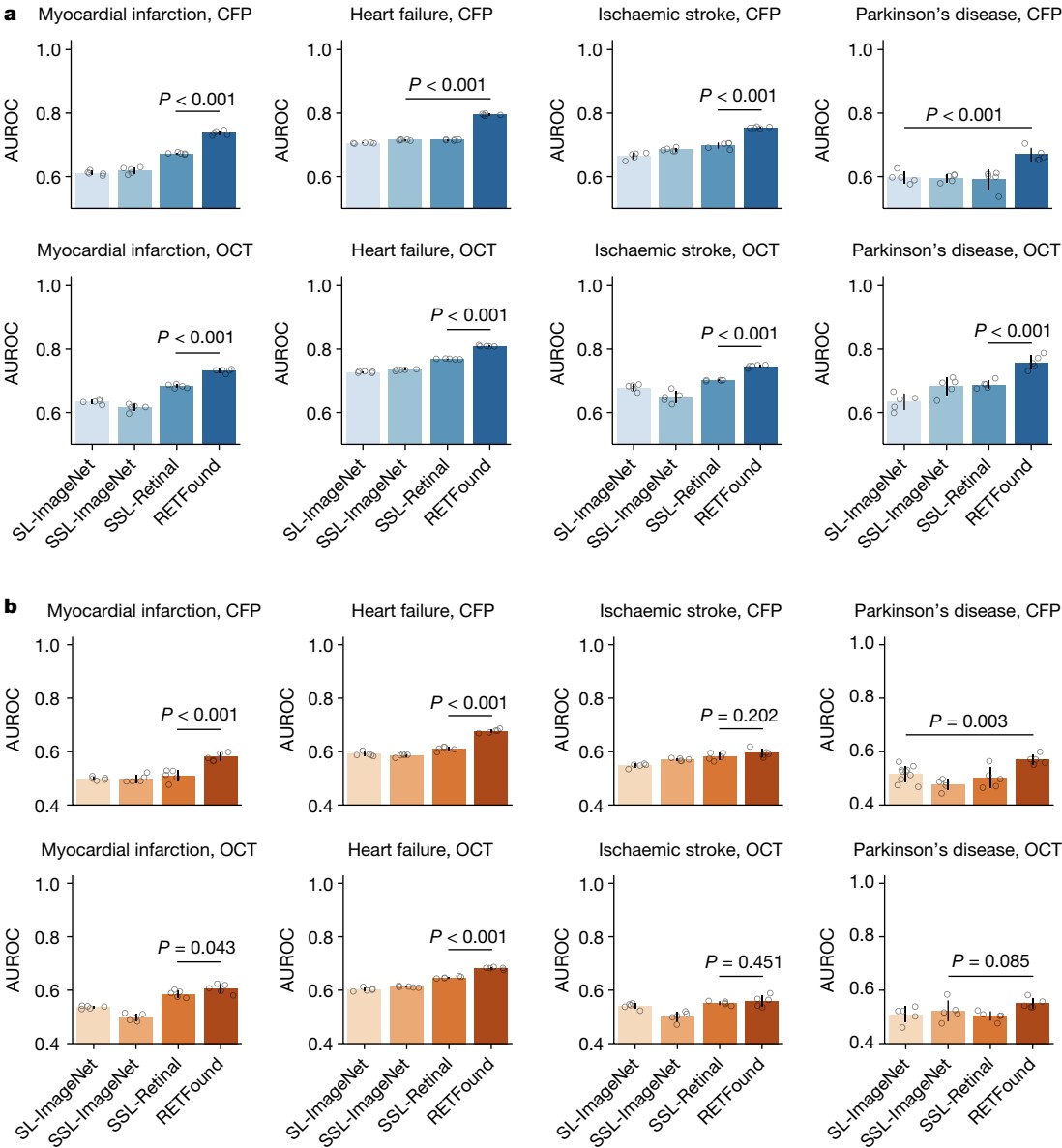

**Fig. 3 | Performance on 3-year incidence prediction of systemic diseases with retinal images. a**, Internal evaluation. Models are adapted to curated datasets from MEH-AlzEye by fine-tuning and internally evaluated on hold-out test data. **b**, External evaluation. Models are fine-tuned on MEH-AlzEye and externally evaluated on the UK Biobank. Data for internal and external evaluation are described in Supplementary Table 2. Although the overall performances are not high due to the difficulty of tasks, RETFound achieved significantly higher AUROC in all internal evaluations and most external evaluations. For each task, we trained the model with five different random seeds, determining the shuffling of training data, and evaluated the models on the test set to get five replicas. We derived the statistics with the five replicas. The error bars show 95% CI and the bar centre represents the mean value of the AUROC. We compare the performance of RETFound with the most competitive comparison model to check whether statistically significant differences exist. *P* value is calculated with the two-sided *t*-test and listed in the figure.

that anatomical structures associated with systemic conditions, such as the optic nerve on CFP and nerve fibre layer and ganglion cell layer on OCT, were highlighted as areas that contributed to the incidence prediction of systemic diseases (Extended Data Fig. 6b).

## Robustness to age distribution shifts

For ageing-associated systemic diseases, clinically relevant anatomical structures alter with both ageing[43,44] and disease progression[19,20,22]. RETFound was trained to identify general structure alterations for detection of systemic diseases (Extended Data Fig. 6b). To further verify the extent to which models can learn anatomical structure changes, respectively, relating to ageing and disease progression, we evaluated performance of the models when using four different control groups with varying ages (mean ages 66.8, 68.5, 70.4 and 71.9 years) versus a fixed disease group (mean age 72.1 years) in the task of myocardial infarction. As shown in Extended Data Fig. 7, the models showed better performance when the age difference is larger, indicating that age is indeed a confounder for studying ageing-associated diseases. The contribution of age can be demonstrated by the extreme case in which the age difference between cohorts is maximal (5.3 years in our scenario), at which point a simple logistic regression with the input of age achieved an AUROC of 0.63, surpassing SSL-ImageNet and SL-ImageNet. When the age difference decreased, the models clearly outperformed the logistic regression. We observed that RETFound kept stable performance even when the age difference decreased, which suggested that RETFound well identified the disease-related anatomical structure alteration and used the information for predicting systemic diseases.

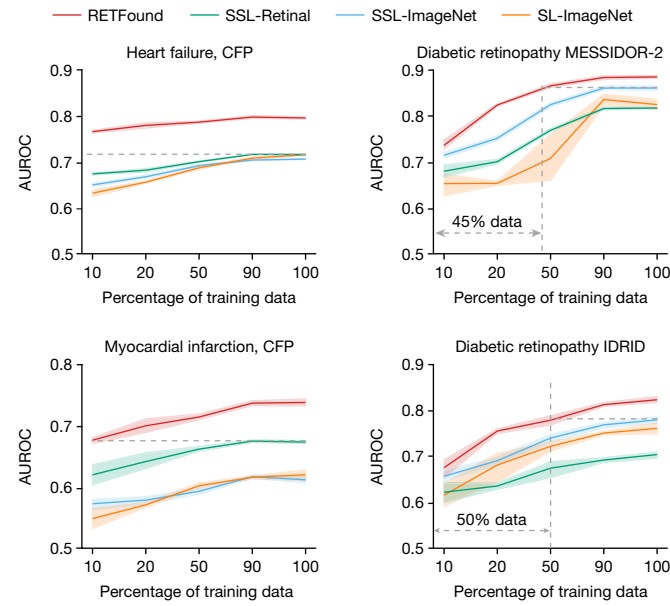

**Fig. 4 | Label efficiency in exemplary applications.** Label efficiency measures the performance with different fractions of training data to understand the amount of data required to achieve a target performance level. The dashed grey lines highlight the difference in training data between RETFound and the most competitive comparison model. RETFound performs better than the comparison groups with 10% of training data in 3-year incidence prediction of heart failure and myocardial infarction with modality of CFP and comparable to other groups with 45% of data in diabetic retinopathy MESSIDOR-2 and 50% of data on IDRID. The 95% CI of AUROC are plotted in colour bands and the centre points of the bands indicate the mean value of AUROC.

## Discussion

This work introduces a new SSL-based foundation model, RETFound, and evaluates its generalizability in adapting to diverse downstream tasks. After training on large-scale unlabelled retinal images using an advanced SSL technique (masked autoencoder), RETFound can be efficiently adapted to a broad range of disease detection tasks, resulting in significant performance improvements for detecting ocular diseases and predicting cardiovascular and neurodegenerative diseases. It is a medical foundation model that has been developed and assessed, and shows considerable promise for leveraging such multidimensional data without constraints of enormous high-quality labels.

RETFound enhances the performance of detecting ocular diseases by learning to identify disease-related lesions. Ocular diseases are diagnosed by the presence of well-defined pathological patterns, such as hard exudates and haemorrhages for diabetic retinopathy. These features involve abnormal variations in colour or structure, showing visible differences from the surrounding retina. RETFound can identify disease-related patterns and correctly diagnose ocular diseases (for example, myopia and diabetic retinopathy cases in Extended Data Fig. 6b). In Fig. 2, we observe that RETFound ranks first in various tasks, followed by SL-ImageNet. SL-ImageNet pretrains the model using supervised learning on ImageNet-21k, which contains 14 million images with 21,000 categories of natural objects with diverse shapes and textures, such as zebras and oranges. Such diverse characteristics allow models to learn abundant low-level features (for example, lines, curves and edges) to identify the boundary of abnormal patterns, thus improving disease diagnosis when the model adapts to medical tasks. In this paper, we demonstrate that by using SSL successively on natural images and unlabelled retinal images, a generalizable foundation model (RETFound) can be developed to further improve ocular disease diagnosis and prognosis, even outperforming the powerful SL-ImageNet.

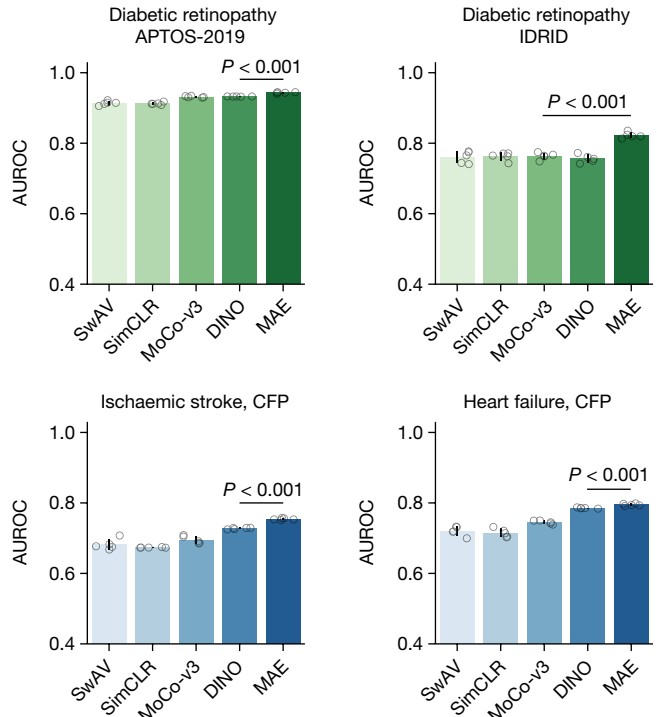

**Fig. 5 | Comparison of different SSL strategies in RETFound framework on exemplar applications.** We show AUROC of predicting diabetic retinopathy, ischaemic stroke and heart failure by the models pretrained with different SSL strategies, including the masked autoencoder (MAE), SwAV, SimCLR, MoCo-v3 and DINO. The data for systemic disease tasks come from the MEH-AlzEye dataset. RETFound with MAE achieved significantly higher AUROC in most tasks. The corresponding quantitative results for the contrastive SSL approaches are listed in Supplementary Table 4. For each task, we trained the model with five different random seeds, determining the shuffling of training data, and evaluated the models on the test set to get five replicas. We derived the statistics with the five replicas. The error bars show 95% CI and the bar centre represents the mean value of the AUPR. We compare the performance of RETFound with the most competitive comparison model to check whether statistically significant differences exist. *P* value is calculated with the two-sided *t*-test and listed in the figure.

RETFound learns retina-specific context by SSL on unlabelled retinal data to improve the prediction of systemic health states. RETFound and SSL-Retinal rank top 2 in both internal and external evaluation in predicting systemic diseases by using SSL on unlabelled retinal images (Fig. 3). In pretraining RETFound learns representations by performing a pretext task involving the reconstruction of an image from its highly masked version, requiring the model to infer masked information with limited visible image patches. Solving such a pretext task in retinal images allows the model to learn retina-specific context, including anatomical structures such as the optic nerve and retinal nerve fibre layer (Extended Data Fig. 6a) that are potential markers in retinal images for neurodegenerative diseases and cardiovascular diseases[17,19,21,45]. The confusion matrix shows that RETFound achieves the highest sensitivity (Extended Data Table 1), indicating that more individuals with a high risk of systemic diseases are identified. The evaluation on oculomic tasks demonstrates the use of retinal images for incidence prediction and risk stratification of systemic diseases, significantly promoted by RETFound.

Compared to SSL-Retinal and SSL-ImageNet, RETFound shows consistently better performance for disease detection (Figs. 2 and 3 and Supplementary Table 3), thus demonstrating SSL on retinal and natural images is complementary to developing the powerful foundation model. The strategy of combining natural images and medical data in

model development has also been validated in other medical fields, such as chest X-rays[6] and dermatology imaging[46]. We also conducted calibration analyses for prediction models in oculomic tasks, which examines the agreement between predicted probabilities and real incidence. A well-calibrated model can provide a meaningful and reliable disease prediction as the predicted probability indicates the real likelihood of disease occurrence, enabling the risk stratification of diseases[47,48]. We observed that RETFound was better calibrated compared to other models and showed the lowest expected calibration error in the reliability diagram (Extended Data Fig. 8). This verifies that RETFound generates reliable predicted probabilities, rather than overconfident ones.

The experiments show that both modalities of CFP and OCT have unique ocular and systemic information encoded that is valuable in predicting future health states. For ocular diseases, some image modalities are commonly used for a diagnosis in which the specific lesions can be well observed, such as OCT for wet-AMD. However, such knowledge is relatively vague in oculomic tasks as (1) the markers for oculomic research on different modalities are under exploration and (2) it requires a fair comparison between many modalities with identical evaluation settings. In this work, we investigate and compare the efficacy of CFP and OCT for oculomic tasks with identical training and evaluation details (for example, train, validation and/or test data splitting is aligned by anonymous patient IDs). We notice that the models with CFP and OCT achieve unequal performances in predicting systemic diseases (Fig. 3 and Supplementary Table 3), suggesting that CFP and OCT contain different levels of information for oculomic tasks. For instance, in 3-year incidence prediction of ischaemic stroke, RETFound with CFP performs better than with OCT on both MEH-AlzEye (internal evaluation) and UK Biobank (external evaluation). For the task of Parkinson's disease, RETFound with OCT shows significantly better performance in internal evaluation. These observations may indicate that various disorders of ageing (for example, stroke and Parkinson's disease) manifest different early markers on retinal images. A practical implication for health service providers and imaging device manufacturers is to recognize that CFP has continuing value, and should be retained as part of the standard retinal assessment in eye health settings. This observation also encourages oculomic research to investigate the strength of association between systemic health with the information contained in several image modalities.

There is a significant fall in performance when adapted models are tested against new cohorts that differ in the demographic profile, and even on the imaging devices that were used (external evaluation phase). This phenomenon is observed both in the external evaluation of ocular disease diagnosis (Fig. 2b) and systemic disease prediction (Fig. 3b). For example, the performance on ischaemic stroke drops (RETFound's AUROC decreases by 0.16 with CFP and 0.19 with OCT). In the challenging oculomic tasks, the age and ethnicity profile of the internal and external validation cohorts (MEH-AlzEye and UK Biobank) as well as the imaging devices are significantly different (Supplementary Table 2), and this is likely to be reflected in the drop in performance when externally evaluated in the UK Biobank cohort. Compared to other models, RETFound achieves significantly higher performance in external evaluation in most tasks (Fig. 3b) as well as different ethnicities (Extended Data Figs. 9–11), showing good generalizability.

We observe that RETFound maintains competitive performance for disease detection tasks, even when substituting various contrastive SSL approaches into the framework (Fig. 5 and Extended Data Fig. 5). It seems that the generative approach using the masked autoencoder generally outperforms the contrastive approaches, including SwAV, SimCLR, MoCo-v3 and DINO. However, it is worth noting that asserting the superiority of the masked autoencoder requires caution, given the presence of several variables across all models, such as network architectures (for example, ResNet-50 for SwAV and SimCLR, Transformers for the others) and hyperparameters (for example, learning rate scheduler). Our comparison demonstrates that the combination of powerful network architecture and complex pretext tasks can produce effective and general-purpose medical foundation models, aligning with the insights derived from large language models in healthcare[49,50]. Furthermore, the comparison further supports the notion that the retinal-specific context learned from the masked autoencoder's pretext task, which includes anatomical structures such as the optic nerve head and retinal nerve fibre layer (as shown in Extended Data Fig. 6a), indeed provides discriminative information for the detection of ocular and systemic diseases.

We believe that research on medical foundation models, such as RETFound, has the potential to democratize access to medical AI and accelerate progress towards widespread clinical implementation. To this end, foundation models must learn powerful representations from enormous volumes of medical data (1.6 million retinal images in our case), which is often only accessible to large institutions with efficient dataset curation workflows. Also, SSL pretraining of foundation models requires many computational resources to achieve training convergence. We used eight NVIDIA Tesla A100 (40 GB) graphical processing units (GPUs) on the Google Cloud Platform, requiring 2 weeks of developing time. By contrast, the data and computational requirements required to fine-tune RETFound to downstream tasks are comparatively small and therefore more achievable for most institutions. We required only one NVIDIA Tesla T4 (16 GB) GPU, requiring about 1.2 h with a dataset of 1,000 images. Moreover, foundational models offer the potential to raise the general quality of healthcare AI models. Their adoption may help avoid superficially impressive models that rarely affect clinical care. These poorly generalizable models consume significant resources and can feed scepticism about the benefits of AI in healthcare. By making RETFound publicly available, we hope to accelerate the progress of AI in medicine by enabling researchers to use our large dataset to design models for use in their own institutions or to explore alternative downstream applications.

Although this work systematically evaluates RETFound in detecting and predicting diverse diseases, there are several limitations and challenges requiring exploration in future work. First, most data used to develop RETFound came from UK cohorts, therefore it is worth exploring the impact of introducing a larger dataset by incorporating retinal images worldwide, with more diverse and balanced data distribution. Second, although we study the performance with modalities of CFP and OCT, the multimodal information fusion between CFP and OCT has not been investigated, which might lead to further improvement in performance. Finally, some clinically relevant information, such as demographics and visual acuity that may work as potent covariates for ocular and oculomic research, has not been included in SSL models. Combining these, we propose to further enhance the strength of RETFound in subsequent iterations by introducing even larger quantities of images, exploring further modalities and enabling dynamic interaction across multimodal data. While we are optimistic about the broad scope of RETFound to be used for a range of AI tasks, we also acknowledge that enhanced human–AI integration is critical to achieving true diversity in healthcare AI applications.

In conclusion, we have verified the efficacy and efficiency of RETFound in adapting to diverse healthcare applications, showing high performance and generalizability in detecting ocular diseases and significant improvement in predicting systemic diseases. By overcoming current barriers to clinical AI applications—notably, the extent of labelled data and limited performance and generalizability—SSL-based foundation models open the door to accelerated, data-efficient devices that may transform care for patients with ocular or systemic diseases.

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

**UK Biobank Eye & Vision Consortium**

**Naomi Allen[13], John E. J. Gallacher[13], Thomas Littlejohns[13], Tariq Aslam[14], Paul Bishop[14], Graeme Black[14], Panagiotis Sergouniotis[14], Denize Atan[15], Andrew D. Dick[15], Cathy Williams[15], Sarah Barman[16], Jenny H. Barrett[17], Sarah Mackie[17], Tasanee Braithwaite[18], Roxana O. Carare[19], Sarah Ennis[19], Jane Gibson[19], Andrew J. Lotery[19], Jay Self[19], Usha Chakravarthy[20], Ruth E. Hogg[20], Euan Paterson[20], Jayne Woodside[20], Tunde Peto[20], Gareth Mckay[20], Bernadette Mcguinness[20], Paul J. Foster[2,4], Konstantinos Balaskas[2,4], Pearse A. Keane[2,4], Anthony P. Khawaja[2,4], Nikolas Pontikos[2,4], Jugnoo S. Rahi[2,4], Gerassimos Lascaratos[2], Praveen J. Patel[2], Michelle Chan[2], Sharon Y. L. Chua[2], Alexander Day[2], Parul Desai[2], Cathy Egan[2], Marcus Fruttiger[2], David F. Garway-Heath[2], Alison Hardcastle[2], Sir Peng T. Khaw[2], Tony Moore[2], Sobha Sivaprasad[2], Nicholas Strouthidis[2], Dhanes Thomas[2], Adnan Tufail[2], Ananth C. Viswanathan[2], Alastair K. Denniston[10,11], Bal Dhillon[21], Tom Macgillivray[21], Cathie Sudlow[21], Veronique Vitart[21], Alexander Doney[22], Emanuele Trucco[22], Jeremy A. Guggenheim[23], James E. Morgan[23], Chris J. Hammond[24], Katie Williams[24], Pirro Hysi[24], Simon P. Harding[25], Yalin Zheng[25], Robert Luben[4], Phil Luthert[4], Zihan Sun[4], Martin McKibbin[26], Eoin O'Sullivan[27], Richard Oram[28], Mike Weedon[28], Chris G. Owen[29], Alicja R. Rudnicka[29], Naveed Sattar[30], David Steel[31], Irene Stratton[32], Robyn Tapp[33], Max M. Yates[34], Axel Petzold[35] & Savita Madhusudhan[36]**

[13]University of Oxford, Oxford, UK. [14]University of Manchester, Manchester, UK. [15]University of Bristol, Bristol, UK. [16]Kingston University, London, UK. [17]University of Leeds, Leeds, UK. [18]St Thomas' Hospital, London, UK. [19]University of Southampton, Southampton, UK. [20]Queens University Belfast, Belfast, UK. [21]University of Edinburgh, Edinburgh, UK. [22]University of Dundee, Dundee, UK. [23]Cardiff University, Cardiff, UK. [24]King's College London, London, UK. [25]University of Liverpool, Liverpool, UK. [26]Leeds Teaching Hospitals NHS Trust, Leeds, UK. [27]King's College Hospital NHS Foundation Trust, London, UK. [28]University of Exeter, Exeter, UK. [29]University of London, London, UK. [30]University of Glasgow, Glasgow, UK. [31]Newcastle University, Newcastle, UK. [32]Gloucestershire Hospitals NHS Foundation Trust, Gloucester, UK. [33]St George's University of London, London, UK. [34]University of East Anglia, Norwich, UK. [35]UCL Institute of Neurology, London, UK. [36]Royal Liverpool University Hospital, Liverpool, UK.

## Methods

### Datasets for developing RETFound

We curate large collections of unannotated retinal images for SSL, totalling 904,170 CFPs and 736,442 OCT scans. Of these, 815,468 (90.2%) CFPs and 627,133 (85.2%) OCTs are from Moorfields Diabetic imAge dataSet (MEH-MIDAS), and 88,702 (9.8%) CFPs are Kaggle EyePACS and 109,309 (14.8%) OCTs that come from ref. 34. MEH-MIDAS is a retrospective dataset that includes the complete ocular imaging records of 37,401 patients (16,429 female, 20,966 male and six unknown) with diabetes who were seen at Moorfields Eye Hospital, London, UK between 2000 and 2022. The age distribution has a mean value of 64.5 and standard deviation of 13.3. The ethnicity distributes diversely: British (13.7%), Indian (14.9%), Caribbean (5.2%), African (3.9%), other ethnicity (37.9%) and not stated (24.4%). MEH-MIDAS includes various imaging devices, such as topcon 3DOCT-2000SA (Topcon), CLARUS (ZEISS) and Triton (Topcon). EyePACS includes images devices of Centervue DRS (Centervue), Optovue iCam (Optovue), Canon CR1/DGi/CR2 (Canon) and Topcon NW (Topcon). Reference 34 contains images from SPECTRALIS (Heidelberg).

### Data for ocular disease diagnosis

We evaluate the model performance on three different categories of disease detection tasks. The first category of tasks involves diagnostic classification of ocular diseases with publicly available ophthalmic data. For diabetic retinopathy diagnosis, Kaggle APTOS-2019 (India), IDRID (India) and MESSIDOR-2 (France) are used. The labels for diabetic retinopathy are based on the International Clinical Diabetic Retinopathy Severity scale, indicating five stages from no diabetic retinopathy to proliferative diabetic retinopathy. For glaucoma, PAPILA[51] (Spain) and Glaucoma Fundus[52] (South Korea) are included. Glaucoma Fundus and PAPILA have three categorical labels, non-glaucoma, early glaucoma (suspected glaucoma) and advanced glaucoma. For datasets with several diseases, JSIEC[53] (China), Retina and OCTID[54] (India) are included. JSIEC includes 1,000 images with 39 categories of common referable fundus diseases and conditions. Retina has labels of normal, glaucoma, cataract and retina disease. OCTID includes 470 OCT scans with labels of normal, macular hole, AMD, central serous retinopathy and diabetic retinopathy. The grading protocols for the public datasets are summarized as: IDRiD, two medical experts provided adjudicated consensus grades; MESSIDOR-2, adjudicated by a panel of three retina specialists in accordance with a published protocol[55]; APTOS-2019, Kaggle dataset with limited information but possibly a single clinician grader; PAPILA, labelling and segmentation by two experts following extensive clinical examination and testing procedure including a retrospective clinical record review; Glaucoma Fundus, agreement of two specialists based on visual fields and extensive imaging and JSIEC, labelled by ophthalmologists and confirmed by senior retina specialists. Disagreements resolved by panel of five senior retina specialists were as follows: Retina, details not available and OCTID, describes image labelling based on the diagnosis of retinal clinical experts but does not specify duplicate adjudication. The details of datasets, such as imaging devices, country and label category, are listed in Supplementary Table 1.

### Data for disease prognosis and prediction

For disease prognosis of fellow eye converting to wet-AMD in 1 year, we use data from the Moorfields AlzEye study (MEH-AlzEye). MEH-AlzEye is a retrospective cohort study linking ophthalmic data of 353,157 patients, who attended Moorfields Eye Hospital between 2008 and 2018, with systemic health data from hospital admissions across the whole of England. Systemic health data are derived from Hospital Episode Statistics (HES) data relating to admitted patient care, with a focus on cardiovascular disease and all-cause dementia. Diagnostic codes in HES admitted patient care are reported according to the tenth revision of the ICD (International Statistical Classification of Diseases)[56].

In line with previous reports, we selected the study cohort using ICD code: stroke (I23-I24), myocardial infarction (I21-I22), heart failure (I50) and Parkinson's disease (G20). Among 186,651 patients with HES, 6,504 patients are diagnosed with wet-AMD in at least one eye, 819 patients have retinal imaging within 1 year before their fellow eyes convert to wet-AMD and 747 patients with their fellow eyes not converting wet-AMD, after excluding other eye diseases. The final category of tasks studies the 3-year prediction of systemic diseases, with a focus on cardiovascular and neurodegenerative dysfunctions, using the MEH-AlzEye and UK Biobank. The UK Biobank includes 502,665 UK residents aged between 40 and 69 years who are registered with the National Health Service. Among all participants, 82,885 get CFP and OCT examinations and a total of 171,500 retinal images are collected. For each patient, we only include the retinal image from the left eye in one visit, to avoid potential bias by inconsistent individual visits. For internal evaluation, we split the patient groups into training, validation and test sets at a ratio of 55:15:30%. The training set is used to revise model parameters to achieve objective function. The validation set is for monitoring training converge and checkpoint selection. The test set is used to test the saved model checkpoint and evaluate the internal performance. For external validation, all patient data are used for evaluating the saved model checkpoint. The detailed data flowcharts are listed in Supplementary Figs. 1–5.

### Data processing and augmentation for SSL

For CFP image preprocessing, we use AutoMorph[57], an automated retinal image analysis tool, to exclude the background and keep the retinal area. All images are resized to 256 × 256 with cubic interpolation. For OCT, we extract the middle slices and resize them to 256 × 256. We follow the same data augmentation as the masked autoencoder in model training, including random crop (lower bounds 20% of the whole image and upper bounds 100%) and resizing the cropped patches to 224 × 224, random horizontal flipping and image normalization.

### RETFound architecture and implementation

We use a specific configuration of the masked autoencoder[15], which consists of an encoder and a decoder. The architecture detail is shown in Supplementary Fig. 6. The encoder uses a large vision Transformer[58] (ViT-large) with 24 Transformer blocks and an embedding vector size of 1,024, whereas the decoder is a small vision Transformer (Vit-small) with eight Transformer blocks and an embedding vector size of 512. The encoder takes unmasked patches (patch size of 16 × 16) as input and projects it into a feature vector with a size of 1,024. The 24 Transformer blocks, comprising multiheaded self-attention and multilayer perceptron, take feature vectors as input and generate high-level features. The decoder inserts masked dummy patches into extracted high-level features as the model input and then reconstructs the image patch after a linear projection. In model training, the objective is to reconstruct retinal images from the highly masked version, with a mask ratio of 0.75 for CFP and 0.85 for OCT. The batch size is 1,792 (8 GPUs × 224 per GPU). The total training epoch is 800 and the first 15 epochs are for learning rate warming up (from 0 to a learning rate of $1 \times 10^{-3}$). The model weights at the final epoch are saved as the checkpoint for adapting to downstream tasks.

### Adaptation to downstream tasks

In adapting to downstream tasks, we only need the encoder (ViT-large) of the foundation model and discard the decoder. The encoder generates high-level features from retinal images. A multilayer perceptron takes the features as input and outputs the probability of disease categories. The category with the highest probability will be defined as the final classification. The number of categories decides the neuron of the final layer of the multilayer perceptron. We include label smoothing to regulate the output distribution thus preventing overfitting of the model by softening the ground-truth labels

in the training data. The training objective is to generate the same categorical output as the label. The batch size is 16. The total training epoch is 50 and the first ten epochs are for learning rate warming up (from 0 to a learning rate of $5 \times 10^{-4}$), followed by a cosine annealing schedule (from learning rates of $5 \times 10^{-4}$ to $1 \times 10^{-6}$ in the rest of the 40 epochs). After each epoch training, the model will be evaluated on the validation set. The model weights with the highest AUROC on the validation set will be saved as the model checkpoint for internal and external evaluation.

### Contrastive SSL implementation
We replace the primary SSL approach (that is, masked autoencoder) with SimCLR[16], SwAV[37], DINO[38] and MoCo-v3 (ref. 14) in the RETFound framework to produce variants of the pretrained model for comparison. For SSL training with each contrastive learning approach, we follow the recommended network architectures and hyperparameter settings from the published papers for optimal performance. We first load the pretrained weights on ImageNet-1k to the models and further train the models with 1.6 million retinal images with each contrastive learning approach to obtain pretrained models. We then follow the identical process of transferring the masked autoencoder to fine-tune those pretrained models for the downstream disease detection tasks.

### Explanations for fine-tuned models
We use RELPROP[42] specified for Transformer-based networks. The method uses layer-wise relevance propagation to compute relevancy scores for each attention head in each layer and then integrates them throughout the attention graph, by combining relevancy and gradient information. As a result, it visualizes the areas of input images that lead to a certain classification. RELPROP has been shown to outperform other well-known explanation techniques, such as GradCam[59].

### Computational resources
SSL typically benefits from a large batch size for training and extracting context from data, which requires powerful GPUs for computation. We use eight NVIDIA Tesla A100 (40 GB) on the Google Cloud Platform. It takes about 14 days to develop RETFound. We allocate an equal computational cost to each SSL approach for pretraining. For fine-tuning RETFound to downstream tasks, we use NVIDIA Tesla T4 (16 GB). Fine-tuning takes about 70 min for every 1,000 images.

### Evaluation and statistical analysis
All task performances are evaluated by the classification metrics known as AUROC and AUPR, computed from the receiver operating characteristics and precision-recall curves of classifiers, respectively. For ocular prognosis and oculomic prediction tasks, the AUROC and AUPR are calculated in a binary setting. For multiclass classification, such as five-stage diabetic retinopathy and multicategory disease diagnosis, we calculate the AUROC and AUPR for each disease category and then average them to get the general AUROC and AUPR. For each task, we train the model with five different random seeds, determining the shuffling of training data. We calculate the mean and standard deviation of the performance over the five iterations and calculate the standard error by (standard deviation/$\sqrt{5}$). We obtain the 95% CI by means of $1.96 \times$ standard error. We use the two-sided $t$-tests between the performance of RETFound and the most competitive comparison model to show whether significant differences exist.

### Ethics statement
This study involves human participants and was approved by the London-Central Research Ethics Committee (18/LO/1163, approved 1 August 2018), Advanced statistical modelling of multimodal data of genetic and acquired retinal diseases (20/HRA/2158, approved 5 May 2020) and the Confidential Advisory Group for Section 251 support (18/CAG/0111, approved 13 September 2018). The National Health Service Health Research Authority gave final approval on 13 September 2018. Moorfields Eye Hospital NHS Foundation Trust validated the de-identifications. Only de-identified retrospective data were used for research, without the active involvement of patients.

### Reporting summary
Further information on research design is available in the Nature Portfolio Reporting Summary linked to this article.

## Data availability
The MIDAS dataset consists of routinely collected healthcare data. Owing to its sensitive nature and the risk of reidentification, the dataset is subject to controlled access by means of a structured application process. Data access enquiries may be made to enquiries@insight. hdrhub.org and we will aim to respond within 2 weeks. Further details about the data request pipeline may be found on the INSIGHT Health Data Research Hub website https://www.insight.hdrhub.org. The AlzEye dataset is subject to the contractual restrictions of the data sharing agreements between National Health Service Digital, Moorfields Eye Hospital and University College London, and is not available for access beyond the AlzEye research team. National and international collaborations are welcomed, although restrictions on access to the cohort mean that only the AlzEye researchers can directly analyse individual-level systemic health data. More details can be found at https://reading-centre.org/studies/artificial_intelligence/alzeye. UK Biobank data are available at https://www.ukbiobank.ac.uk/. Data for ocular disease experiments are publicly available online and can be accessed through the following links: IDRID (https://ieee-dataport.org/open-access/indian-diabetic-retinopathy-image-dataset-idrid), MESSIDOR-2 (https://www.adcis.net/en/third-party/messidor2/), APTOS-2019 (https://www.kaggle.com/competitions/aptos2019-blindness-detection/data), PAPILA (https://figshare.com/articles/dataset/PAPILA/14798004/1), Glaucoma Fundus (https://dataverse.harvard.edu/dataset.xhtml?persistentId=doi:10.7910/DVN/1YRRAC), JSIEC (https://zenodo.org/record/3477553), Retina (https://www.kaggle.com/datasets/jr2ngb/cataractdataset) and OCTID (https://borealisdata.ca/dataverse/OCTID).

## Code availability
The code used to train, fine-tune and evaluate RETFound from Y.Z. is available at https://github.com/rmaphoh/RETFound_MAE, which is based on PyTorch. Furthermore, a Keras version implemented by Y.K. is available at https://github.com/uw-biomedical-ml/RETFound_MAE. Please note that the reported results are obtained from PyTorch models. Images were processed with automated retinal image analysis tool AutoMorph v.1.0 (https://github.com/rmaphoh/AutoMorph). Image data were extracted from Dicom files with Pydicom v.2.3.0. Results were further analysed and visualized with Python v.3.6, NumPy v.1.19.5, SciPy v.1.5.4, seaborn v.0.12.0, Matplotlib v.3.6.1, pandas v.1.5.0, Scikit-Learn v.1.1.3 and Pillow v.9.2.0. Heatmaps were generated with RELPROP (https://github.com/hila-chefer/Transformer-Explainability).

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

**Acknowledgements** We thank P. Rawlinson for project management, C. Green and L. Wickham for information governance expertise, and A. Wenban, S. St John-Green and M. Barnfield for information technology support. This work is supported by Engineering and Physical Sciences Research Council grant nos. EP/M020533/1, EP/R014019/1 and EP/V034537/1, as well as the NIHR UCLH Biomedical Research Centre. S.K.W. is supported by a Medical Research Council Clinical Research Training Fellowship (grant no. MR/TR000953/1). P.A.K. is supported by a Moorfields Eye Charity Career Development Award (grant no. R190028A) and a UK Research & Innovation Future Leaders Fellowship (grant no. MR/T019050/1). For the purpose of open access, the author has applied a Creative Commons Attribution (CC BY) licence to any Author Accepted Manuscript version arising.

**Author contributions** Y.Z., M.X., E.J.T., D.C.A. and P.A.K. contributed to the conception and design of the work. Y.Z., M.A.C., S.K.W., D.J.W., R.R.S. and M.G.L. contributed to the data acquisition and organization. Y.Z. contributed to the technical implementation. M.A.C., S.K.W., A.K.D. and P.A.K. provided the clinical inputs to the research. Y.Z., M.A.C., S.K.W., M.S.A., T.L., P.W.-C., A.A., D.C.A. and P.A.K. contributed to the evaluation pipeline of this work. Y.Z., Y.K., A.A., A.Y.L., E.J.T., A.K.D. and D.C.A. provided suggestions on analysis framework. UK Biobank & Eye Vision Consortium provided the UK Biobank. All authors contributed to the drafting and revising of the manuscript.

**Competing interests** P.A.K. has acted as a consultant for DeepMind, Roche, Novartis, Apellis and BitFount, and is an equity owner in Big Picture Medical. He has received speaker fees from Heidelberg Engineering, Topcon, Allergan and Bayer.

**Additional information**
**Correspondence and requests for materials** should be addressed to Yukun Zhou or Pearse A. Keane.

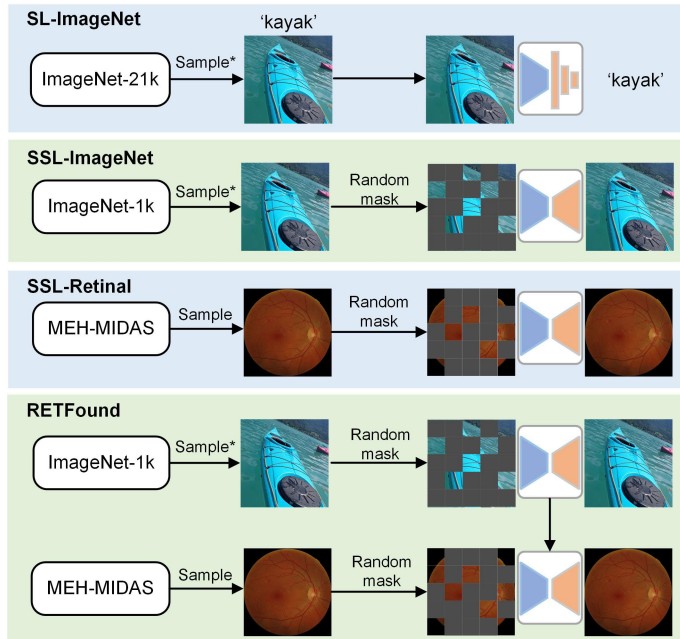

**Extended Data Fig. 1 | Illustration of training pipeline of RETFound and comparison baselines.** The compared baselines include SL-ImageNet, SSL-ImageNet, and SSL-Retinal. SL-ImageNet trains the model via supervised learning on ImageNet-21k (14 million images and categorical labels); SSL-ImageNet trains the model on ImageNet-1k (1.4 million images) via SSL; SSL-Retinal trains the model on retinal images via SSL from scratch; RETFound trains the model on retinal images via SSL from the weights of SSL-ImageNet. *kayak picture is used to illustrate the method pipeline.

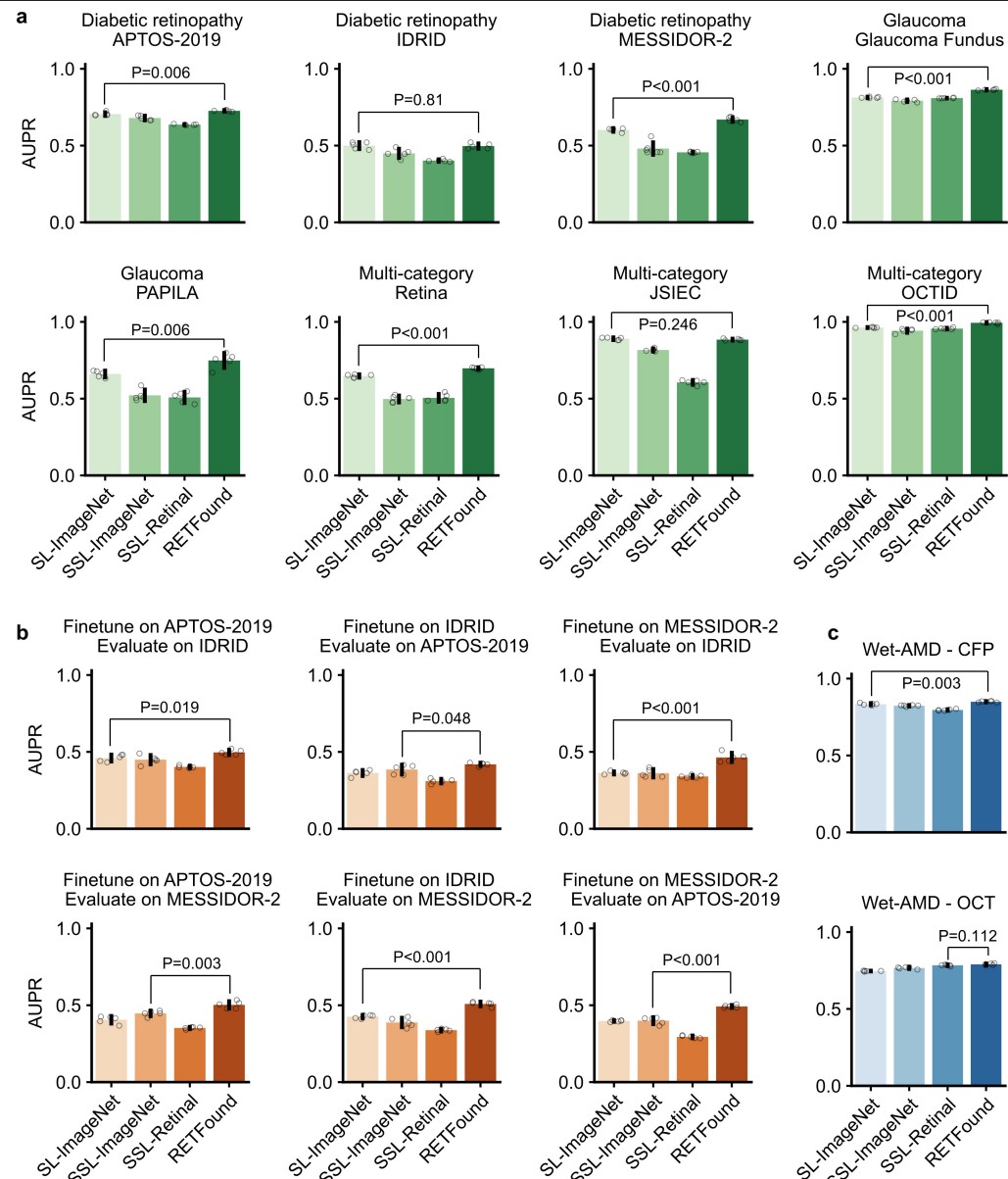

**Extended Data Fig. 2 | Performance (AUPR) on ocular disease diagnostic classification. a**, internal evaluation, models are adapted to each dataset via fine-tuning and internally evaluated on hold-out test data. The dataset details are listed in Supplementary Table 1. **b**, external evaluation, models are fine-tuned on one diabetic retinopathy dataset and externally evaluated on the others. **c**, performance on ocular disease prognosis. The models are fine-tuned to predict the conversion of fellow eye to wet-AMD in 1 year and evaluated internally. For each task, we trained the model with 5 different random seeds, determining the shuffling of training data, and evaluated the models on the test set to get 5 replicas. We derived the statistics with the 5 replicas. The error bars show 95% confidence intervals and the bars' centre represents the mean value of the AUPR. We compare the performance of RETFound with the most competitive comparison model to check if statistically significant differences exist. p-value is calculated with the two-sided t-test and listed in the figure.

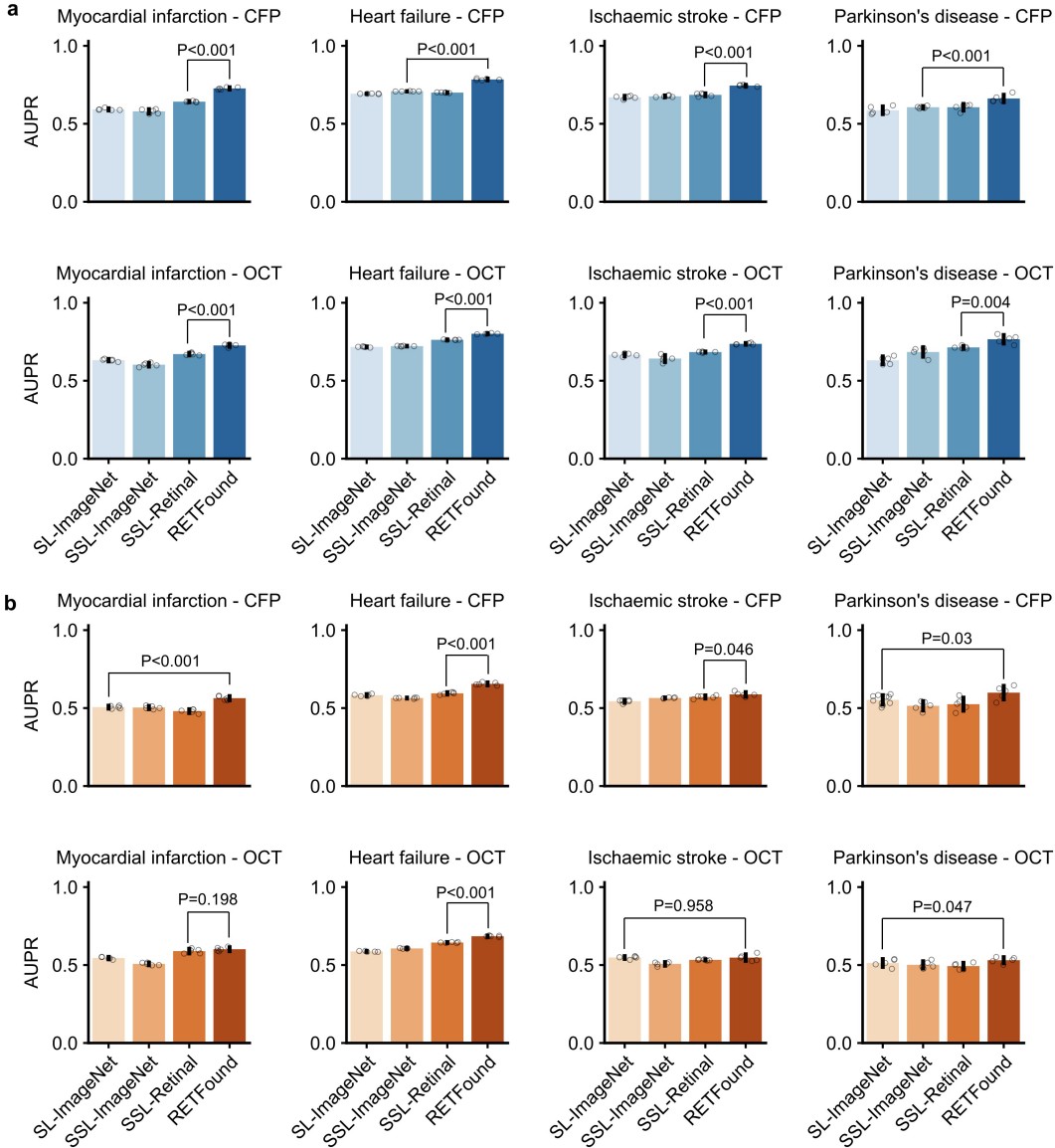

**Extended Data Fig. 3 | Performance (AUPR) on 3-year incidence prediction of systemic diseases with retinal images. a**, internal evaluation, models are adapted to curated datasets from MEH-AlzEye via fine-tuning and internally evaluated on hold-out test data. **b**, external evaluation, models are fine-tuned on MEH-AlzEye and externally evaluated on UK Biobank. Data for internal and external evaluation is described in Supplementary Table 2. For each task, we trained the model with 5 different random seeds, determining the shuffling of training data, and evaluated the models on the test set to get 5 replicas. We derived the statistics with the 5 replicas. The error bars show 95% confidence intervals and the bars' centre represents the mean value of the AUPR. We compare the performance of RETFound with the most competitive comparison model to check if statistically significant differences exist. p-value is calculated with the two-sided t-test and listed in the figure.

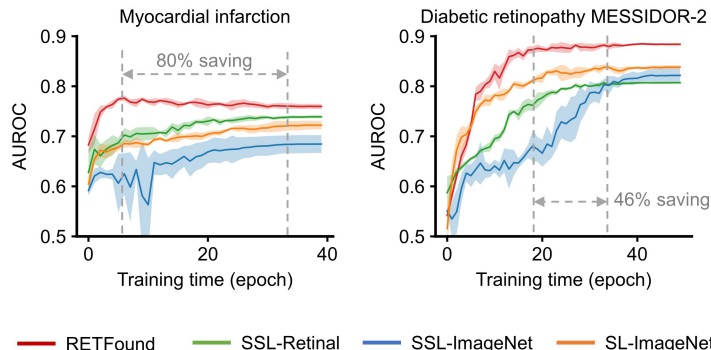

**Extended Data Fig. 4 | Adaptation efficiency in exemplar applications.** Adaptation efficiency refers to the time required to achieve training convergence. We show the performance on validation sets with the same hyperparameters such as learning rate. The gray dash lines highlight the time point when the model checkpoint is saved and the time difference between RETFound and the most competitive comparison model is calculated. RETFound saves 80% of training time in adapting to 3-year incidence prediction of myocardial infarction and 46% in diabetic retinopathy MESSIDOR-2. 95% confidence intervals of AUROC are plotted in colour bands and the mean values are shown as centre lines.

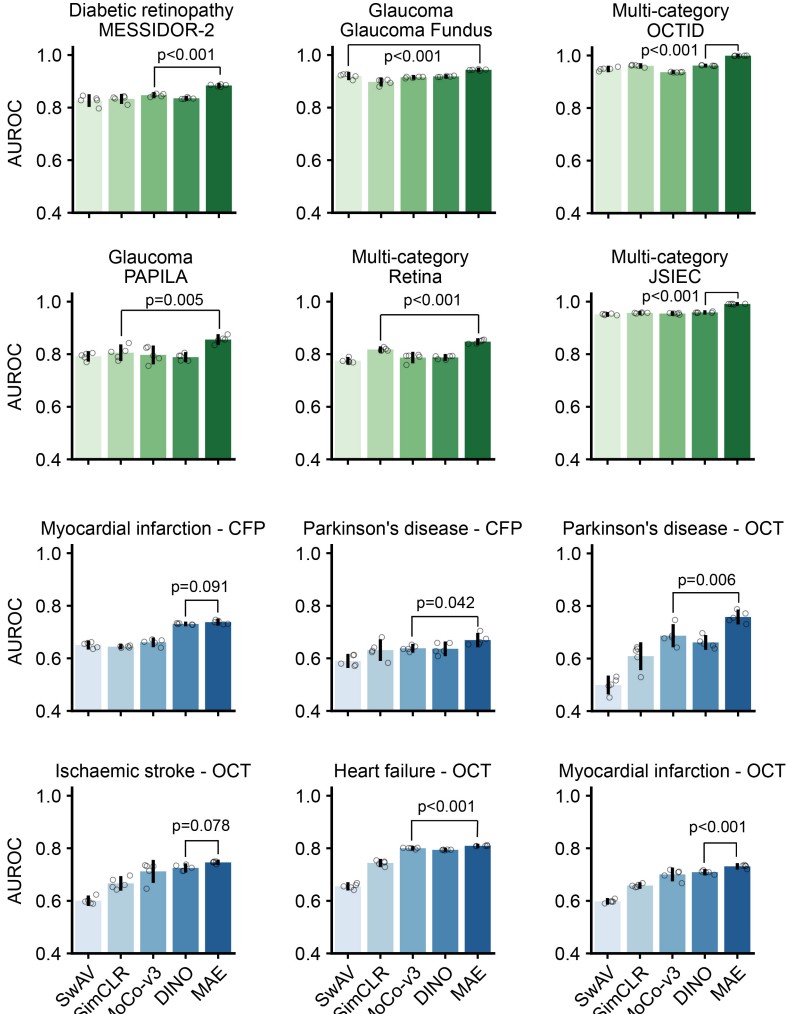

**Extended Data Fig. 5 | Comparison of different SSL strategies in RETFound framework.** We show AUROC of predicting ocular diseases and systemic diseases by the models pretrained with different SSL strategies, including the masked autoencoder (MAE), SwAV, SimCLR, MoCo-v3, and DINO. The corresponding quantitative results for the contrastive SSL approaches are listed in Supplementary Table 4. For each task, we trained the model with 5 different random seeds, determining the shuffling of training data, and evaluated the models on the test set to get 5 replicas. We derived the statistics with the 5 replicas. The error bars show 95% confidence intervals and the bars' centre represents the mean value of the AUPR. We compare the performance of RETFound with the most competitive comparison model to check if statistically significant differences exist. p-value is calculated with the two-sided t-test and listed in the figure.

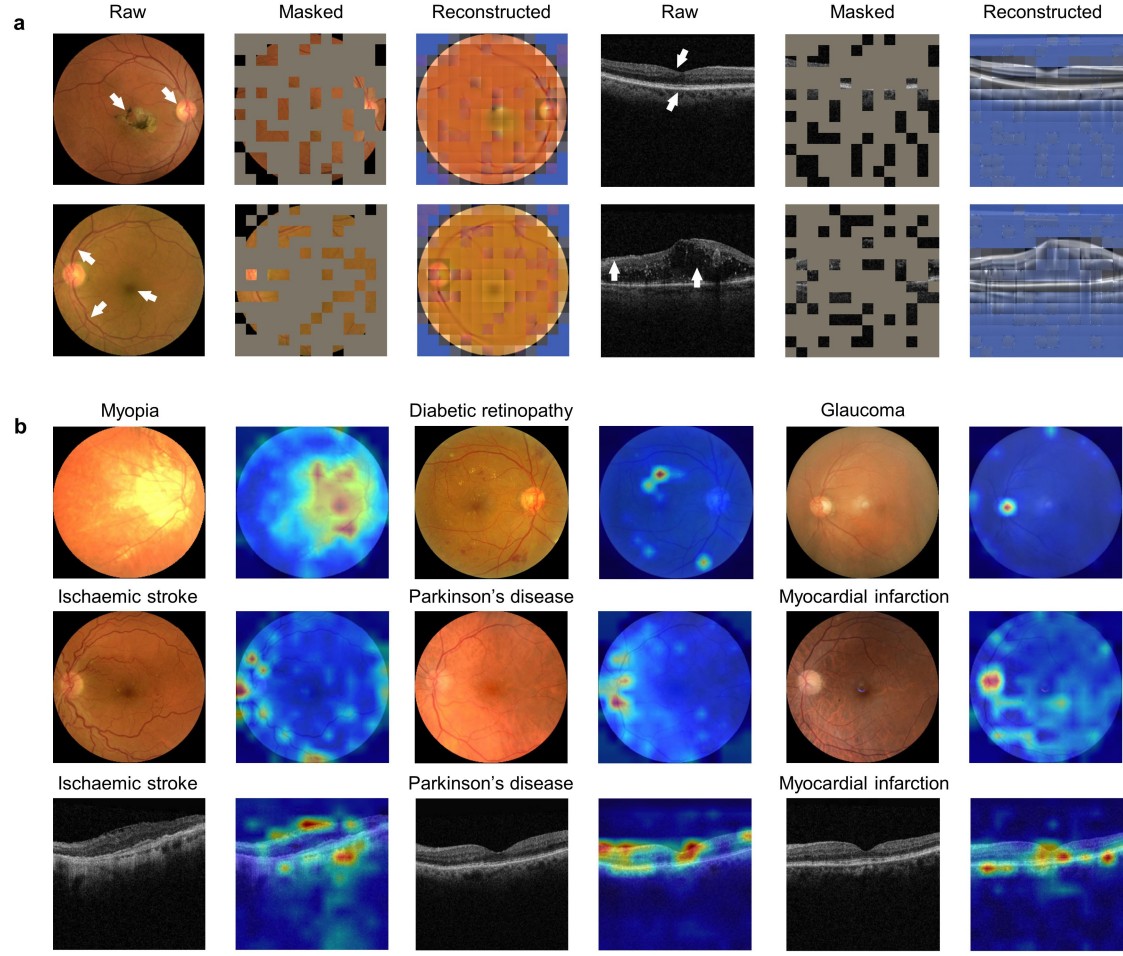

**Extended Data Fig. 6 | Qualitative results of RETFound. a**, Reconstructed colour fundus photographs and optical coherent tomography scans from highly masked images in pretext task. Although with few patches visible, RETFound infers the retina-specific anatomical structures (e.g. optic nerve and retinal nerve fibre layer) and disease lesions, which are markers for multiple diseases. **b**, Heatmaps highlighting the areas that contribute to the classification of the models in various downstream tasks. Red colour indicates high contribution. The well-defined pathologies of ocular diseases are identified and used for classification. For the prediction of systemic diseases, some anatomical structures associated with systemic conditions, e.g. optic nerve and vasculature on CFP and ganglion cell layer and macular area on OCT, are highlighted.

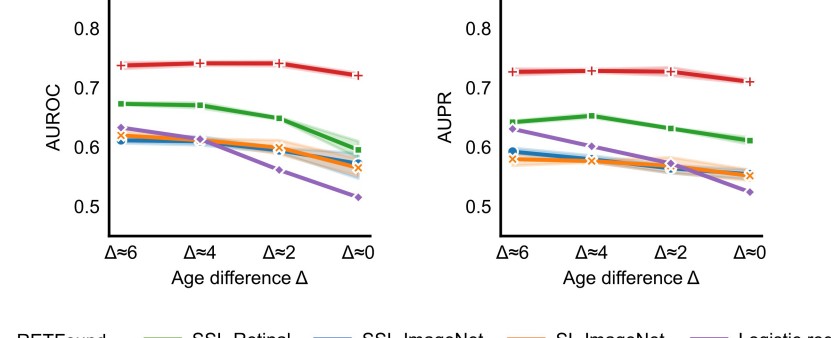

**Extended Data Fig. 7 | Performance on various age distributions in predicting myocardial infarction.** The disease group remains unchanged (mean value of age is 72.1) while the four control groups are sampled with various age distributions (mean values of age are respectively 66.8, 68.5, 70.4, and 71.9). The X axis shows the age difference between disease group and control groups. With each control group, we evaluate the performance of predicting myocardial infarction. The performance of RETFound remains robust to age difference while that of compared models drops when the age difference decreases. Logistic regression uses age as input. The logistic regression performs well when age difference is large (about 6) but clearly worse than SSL models when the difference becomes smaller. 95% confidence intervals are plotted in colour bands and the mean value of performances are shown as the band centres.

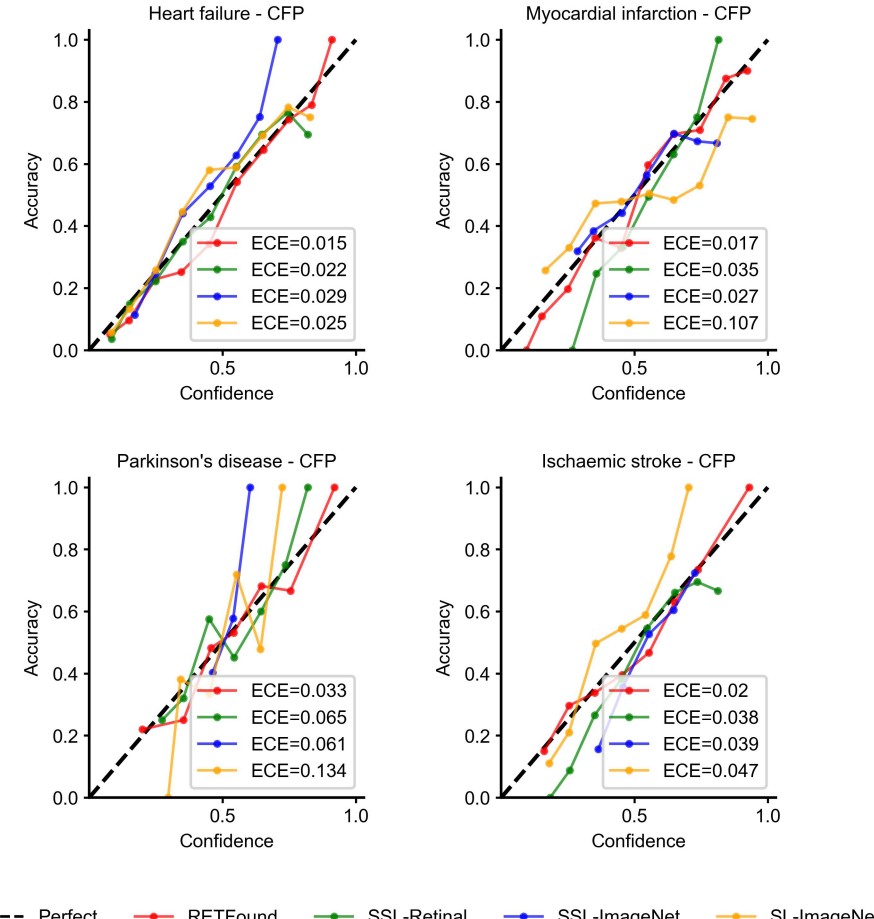

**Extended Data Fig. 8 | Reliability diagrams and expected calibration error (ECE) for prediction models.** Reliability diagrams measure the consistency between the prediction probabilities of an event (e.g. myocardial infarction) with the actual chance of observing the event. The dashed line (diagonal line) indicates a perfectly calibrated model and the deviation represents the miscalibration. RETFound is closest to diagonal lines and the ECE is lowest among all models.

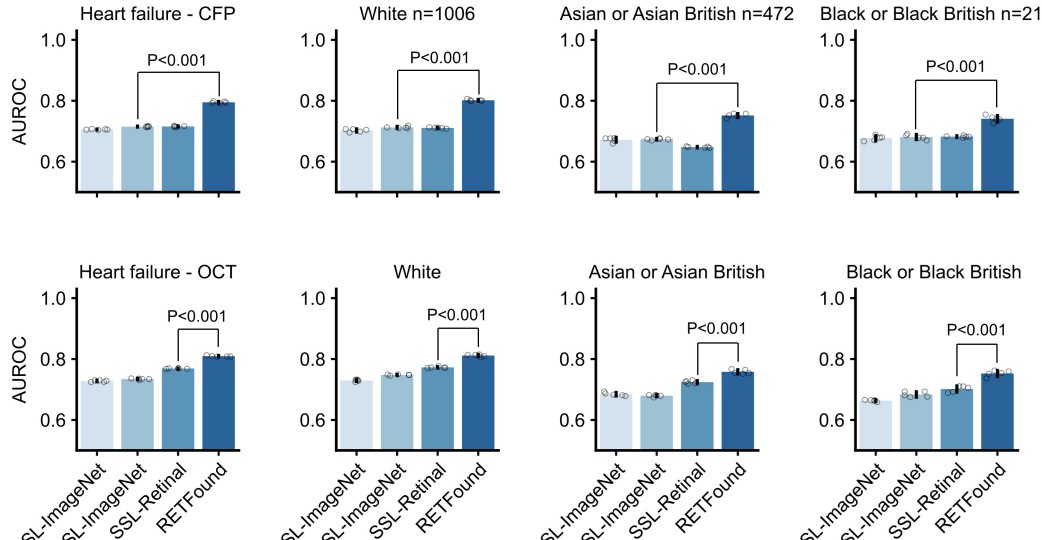

**Extended Data Fig. 9 | Performance in predicting heart failure across ethnicities.** We show AUROC of predicting 3-year heart failure in subsets with different ethnicity, including White, Asian or Asian British, and Black or Black British subgroups, the three largest major categories of ethnicity as described by the UK Government's Office for National Statistics. Data is from MEH-AlzEye test set. The first column shows the performance on all test data, followed by results on three subgroups. The cohort quantity is listed in titles. We trained the model with 5 different random seeds, determining the shuffling of training data, and evaluated the models on the test set to get 5 replicas. We derived the statistics with the 5 replicas. The error bars show 95% confidence intervals and the bars' centre represents the mean value of the AUPR. We compare the performance of RETFound with the most competitive comparison model to check if statistically significant differences exist. p-value is calculated with the two-sided t-test and listed in the figure.

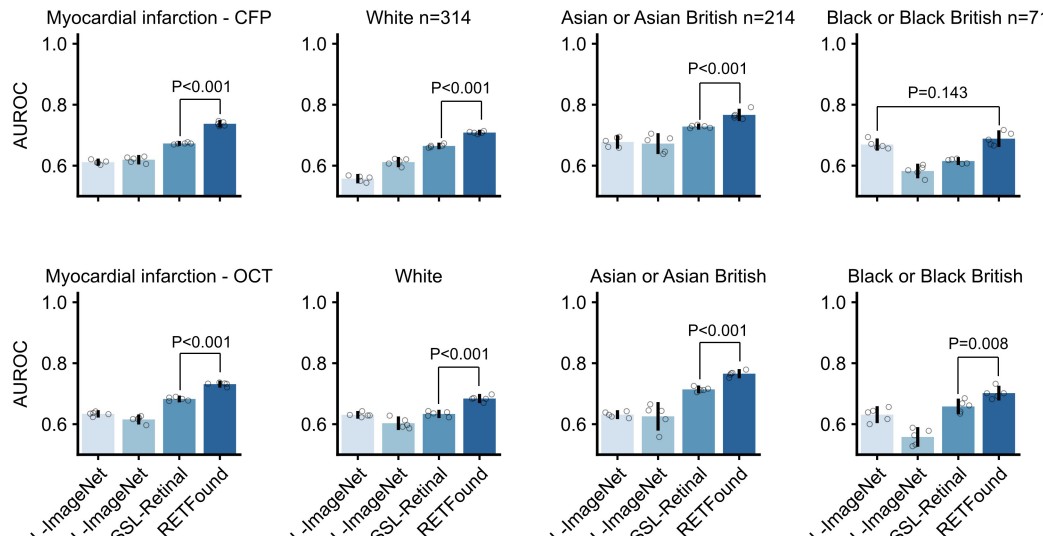

**Extended Data Fig. 10 | Performance in predicting myocardial infarction across ethnicities.** We show AUROC of predicting 3-year myocardial infarction in subsets with different ethnicity. Data is from MEH-AlzEye test set. The first column shows the performance on all test data, followed by results on White, Asian or Asian British, and Black or Black British cohorts. The cohort quantity is listed in titles. We trained the model with 5 different random seeds, determining the shuffling of training data, and evaluated the models on the test set to get 5 replicas. We derived the statistics with the 5 replicas. The error bars show 95% confidence intervals and the bars' centre represents the mean value of the AUPR. We compare the performance of RETFound with the most competitive comparison model to check if statistically significant differences exist. p-value is calculated with the two-sided t-test and listed in the figure.

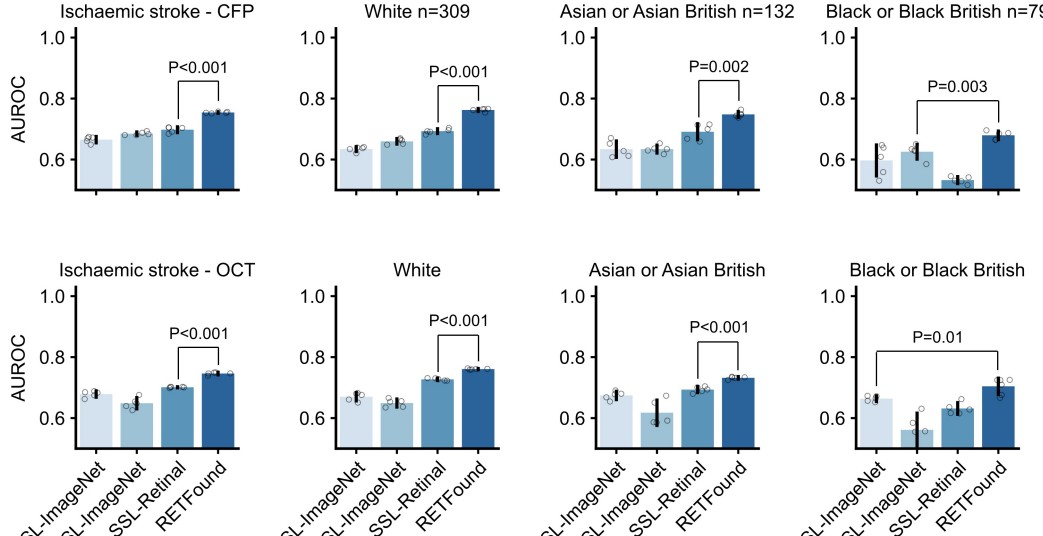

**Extended Data Fig. 11 | Performance in predicting ischaemic stroke across ethnicities.** We show AUROC of predicting 3-year ischaemic stroke in subsets with different ethnicity. Data is from MEH-AlzEye test set. The first column shows the performance on all test data, followed by results on White, Asian or Asian British, and Black or Black British cohorts. The cohort quantity is listed in titles. We trained the model with 5 different random seeds, determining the shuffling of training data, and evaluated the models on the test set to get 5 replicas. We derived the statistics with the 5 replicas. The error bars show 95% confidence intervals and the bars' centre represents the mean value of the AUPR. We compare the performance of RETFound with the most competitive comparison model to check if statistically significant differences exist. p-value is calculated with the two-sided t-test and listed in the figure.

## Extended Data Table 1 | Confusion matrix on 3-year prediction of myocardial infarction

**a**

**SL-ImageNet**

| Label | | Predicted 3-year no MI | Predicted 3-year MI |
|---|---|---|---|
| 3-year no MI | | TN 227 (27%) | FP 189 (23%) |
| 3-year MI | | FN 165 (20%) | TP 251 (30%) |
| Sensitivity: 0.6 | | | Specificity: 0.54 |

**SSL-ImageNet**

| Label | | Predicted 3-year no MI | Predicted 3-year MI |
|---|---|---|---|
| 3-year no MI | | TN 266 (32%) | FP 150 (17%) |
| 3-year MI | | FN 191 (23%) | TP 225 (27%) |
| Sensitivity: 0.54 | | | Specificity: 0.64 |

**SSL-Retinal**

| Label | | Predicted 3-year no MI | Predicted 3-year MI |
|---|---|---|---|
| 3-year no MI | | TN 258 (31%) | FP 158 (19%) |
| 3-year MI | | FN 150 (18%) | TP 266 (32%) |
| Sensitivity: 0.64 | | | Specificity: 0.62 |

**RETFound**

| Label | | Predicted 3-year no MI | Predicted 3-year MI |
|---|---|---|---|
| 3-year no MI | | TN 280 (34%) | FP 136 (16%) |
| 3-year MI | | FN 123 (15%) | TP 293 (35%) |
| Sensitivity: 0.7 | | | Specificity: 0.67 |

**b**

**SL-ImageNet**

| Label | | Predicted 3-year no MI | Predicted 3-year MI |
|---|---|---|---|
| 3-year no MI | | TN 241 (29%) | FP 175 (21%) |
| 3-year MI | | FN 158 (19%) | TP 258 (31%) |
| Sensitivity: 0.62 | | | Specificity: 0.58 |

**SSL-ImageNet**

| Label | | Predicted 3-year no MI | Predicted 3-year MI |
|---|---|---|---|
| 3-year no MI | | TN 258 (31%) | FP 158 (19%) |
| 3-year MI | | FN 200 (24%) | TP 216 (26%) |
| Sensitivity: 0.52 | | | Specificity: 0.62 |

**SSL-Retinal**

| Label | | Predicted 3-year no MI | Predicted 3-year MI |
|---|---|---|---|
| 3-year no MI | | TN 250 (30%) | FP 166 (20%) |
| 3-year MI | | FN 150 (18%) | TP 266 (32%) |
| Sensitivity: 0.64 | | | Specificity: 0.6 |

**RETFound**

| Label | | Predicted 3-year no MI | Predicted 3-year MI |
|---|---|---|---|
| 3-year no MI | | TN 272 (33%) | FP 144 (17%) |
| 3-year MI | | FN 123 (15%) | TP 293 (35%) |
| Sensitivity: 0.7 | | | Specificity: 0.65 |

a, confusion matrix with CFP. b, confusion matrix with OCT. RETFound shows the highest sensitivity and specificity.

# Reporting Summary

## Statistics

For all statistical analyses, confirm that the following items are present in the figure legend, table legend, main text, or Methods section.

| n/a | Confirmed | |
|---|---|---|
| ☐ | ☒ | The exact sample size (*n*) for each experimental group/condition, given as a discrete number and unit of measurement |
| ☐ | ☒ | A statement on whether measurements were taken from distinct samples or whether the same sample was measured repeatedly |
| ☐ | ☒ | The statistical test(s) used AND whether they are one- or two-sided *Only common tests should be described solely by name; describe more complex techniques in the Methods section.* |
| ☒ | ☐ | A description of all covariates tested |
| ☒ | ☐ | A description of any assumptions or corrections, such as tests of normality and adjustment for multiple comparisons |
| ☐ | ☒ | A full description of the statistical parameters including central tendency (e.g. means) or other basic estimates (e.g. regression coefficient) AND variation (e.g. standard deviation) or associated estimates of uncertainty (e.g. confidence intervals) |
| ☐ | ☒ | For null hypothesis testing, the test statistic (e.g. *F*, *t*, *r*) with confidence intervals, effect sizes, degrees of freedom and *P* value noted *Give P values as exact values whenever suitable.* |
| ☒ | ☐ | For Bayesian analysis, information on the choice of priors and Markov chain Monte Carlo settings |
| ☒ | ☐ | For hierarchical and complex designs, identification of the appropriate level for tests and full reporting of outcomes |
| ☒ | ☐ | Estimates of effect sizes (e.g. Cohen's *d*, Pearson's *r*), indicating how they were calculated |

*Our web collection on statistics for biologists contains articles on many of the points above.*

## Software and code

Policy information about availability of computer code

| Data collection | The code used to train, fine-tune, and evaluate RETFound from Yukun Zhou is available at https://github.com/rmaphoh/RETFound_MAE which bases on PyTorch. Additionally, a Keras version implemented by Yuka Kihara is available at https://github.com/uw-biomedical-ml/RETFound_MAE. Please note that the reported results are obtained from PyTorch models. Image data was extracted from Dicom files with Pydicom v2.3.0 (https://pydicom.github.io). Images were processed with automated retinal image analysis tool AutoMorph v1.0 (https://github.com/rmaphoh/AutoMorph). |
|---|---|
| Data analysis | Data was analysed with Python v3.6 (https://www.python.org/), NumPy v1.19.5 (https://github.com/numpy/numpy), SciPy v1.5.4 (https://www.scipy.org/), seaborn v0.12.0 (https://github.com/mwaskom/seaborn), Matplotlib v3.6.1 (https://github.com/matplotlib/matplotlib), pandas v1.5.0 (https://github.com/pandas-dev/pandas), Scikit-Learn v1.1.3 (https://scikit-learn.org/stable), Pillow v9.2.0 (https://pypi.org/project/Pillow). Heatmaps were generated with RELPROP (https://github.com/hila-chefer/Transformer-Explainability). |

For manuscripts utilizing custom algorithms or software that are central to the research but not yet described in published literature, software must be made available to editors and reviewers. We strongly encourage code deposition in a community repository (e.g. GitHub). See the Nature Portfolio guidelines for submitting code & software for further information.

## Data

Policy information about availability of data

All manuscripts must include a data availability statement. This statement should provide the following information, where applicable:

- Accession codes, unique identifiers, or web links for publicly available datasets
- A description of any restrictions on data availability
- For clinical datasets or third party data, please ensure that the statement adheres to our policy

The MIDAS dataset consists of routinely collected healthcare data. Due to its sensitive nature and the risk of reidentification, the dataset is subject to controlled access via a structured application process. Data access enquiries may be made to enquiries@insight.hdrhub.org and we will aim to respond within two weeks. Further details about the data request pipeline may be found on the INSIGHT Health Data Research Hub website https://www.insight.hdrhub.org. The AlzEye dataset is subject to the contractual restrictions of the data sharing agreements between National Health Service Digital, Moorfields Eye Hospital and University College London and are not available for access beyond the AlzEye research team. National and international collaborations are welcomed though restrictions on access to the cohort mean that only the AlzEye researchers can directly analyse individual-level systemic health data. More details can be found at https://readingcentre.org/studies/artificial_intelligence/alzeye. UK Biobank data is available at https://www.ukbiobank.ac.uk/.

Data for ocular disease experiments are publicly available online and can be accessed via the links: IDRID (https://ieee-dataport.org/open-access/indian-diabetic-retinopathy-image-dataset-idrid), MESSIDOR-2 (https://www.adcis.net/en/third-party/messidor2/), APTOS-2019 (https://www.kaggle.com/competitions/aptos2019-blindness-detection/data), PAPILA (https://figshare.com/articles/dataset/PAPILA/14798004/1), Glaucoma Fundus (https://dataverse.harvard.edu/dataset.xhtml?persistentId=doi:10.7910/DVN/1YRRAC), JSIEC (https://zenodo.org/record/3477553), Retina (https://www.kaggle.com/datasets/jr2ngb/cataractdataset), OCTID (https://borealisdata.ca/dataverse/OCTID).

## Human research participants

Policy information about studies involving human research participants and Sex and Gender in Research.

| | |
|---|---|
| Reporting on sex and gender | Biological sex information for MEH-MIDAS and MEH-AlzEye was collected via self-report. MEH-MIDAS includes 37,401 patients (16,429 female, 20,966 male, and 6 unknown) and MEH-AlzEye includes 353,157 patients (190,494 female and 162,663 male). Experiments were conducted both on female and male. We used all MEH-MIDAS data to develop RETFound models and subsets of MEH-AlzEye for downstream validation (detailed in Supplementary Table 2). |
| Population characteristics | MEH-MIDAS is a retrospective dataset which includes the complete ocular imaging records of 37,401 patients with diabetes who were seen at Moorfields Eye Hospital, London, United Kingdom between 2000 and 2022. The age distribution has a mean value of 64.5 and standard deviation of 13.3. The ethnicity distributes diversly: British (13.7%), Indian (14.9%), Caribbean (5.2%), African (3.9%), other ethnicity (37.9%), not stated (24.4%). MEH-MIDAS includes various imaging devices, such as topcon 3DOCT-2000SA (Topcon), CLARUS (ZEISS), and Triton (Topcon).

MEH-AlzEye is a retrospective cohort study linking ophthalmic data of 353,157 patients, who attended Moorfields Eye Hospital between 2008 and 2018, with systemic health data from hospital admissions across the whole of England. Systemic health data are derived from Hospital Episode Statistics (HES) data relating to admitted patient care (APC), with a focus on cardiovascular disease and all-cause dementia. More details can be found in the method section. Selections of study cohort were shown in Supplementary Figure 2-6 and characteristics were listed in Supplementary Table 2.

The UK Biobank includes 502,665 UK residents aged between 40 and 69 years who are registered with the National Health Service. Among all participants, 82,885 get CFP and OCT examinations and a total of 171,500 retinal images are collected. Selections of study cohort were shown in Supplementary Figure 2-6 and characteristics were listed in Supplementary Table 2. |
| Recruitment | MEH-MIDAS is a retrospective dataset which includes the complete ocular imaging records of 37,401 patients with diabetes who were seen at Moorfields Eye Hospital, London, United Kingdom between 2000 and 2022. MEH-AlzEye is a retrospective cohort study linking ophthalmic data of 353,157 patients who attended Moorfields Eye Hospital between 2008 and 2018. |
| Ethics oversight | This study involves human participants and was approved by the London-Central Research Ethics Committee (18/LO/1163, approved 01/08/2018), Advanced statistical modelling of multimodal data of genetic and acquired retinal diseases (20/HRA/2158, approved 05/05/2020), and the Confidential Advisory Group for Section 251 support (18/CAG/0111, approved 13/09/2018). The National Health Service Health Research Authority gave final approval on 13 September 2018. Moorfields Eye Hospital NHS Foundation Trust validated the de-identifications. Only de-identified retrospective data was used for research, without the active involvement of patients. |

Note that full information on the approval of the study protocol must also be provided in the manuscript.

# Field-specific reporting

Please select the one below that is the best fit for your research. If you are not sure, read the appropriate sections before making your selection.

☒ Life sciences          ☐ Behavioural & social sciences          ☐ Ecological, evolutionary & environmental sciences

For a reference copy of the document with all sections, see nature.com/documents/nr-reporting-summary-flat.pdf

# Life sciences study design

All studies must disclose on these points even when the disclosure is negative.

| | |
|---|---|
| Sample size | Data for developing RETFound model was from Moorfields Diabetic imAge dataSet (MEH-MIDAS) and public data (totalling 904,170 CFPs and 736,442 OCTs). Data for ocular disease diagnosis were from public datasets, detailed in Supplementary Table 1. Data for systemic disease prediction were from Moorfields AlzEye project and selected cohorts were introduced in Supplementary Table 2. Datasets were chosen based on the availability of labels that would permit external validation of the different fine-tuned RETFound models, which is dependent on the specific clinical task being evaluated. The chosen external validation datasets were deemed to be suitable based on their parameters, which are summarised Supplementary Information Table 1 Dataset characteristics. Formal sample size calculations were not performed due to the lack of established methods when applied to machine-learning classification studies. |
| Data exclusions | Data failed image processing with AutoMorph were excluded. Data without systemic health labels were excluded. For more details please refer to the method section. |
| Replication | All patients were randomly selected and were not correlated in any way. The replication of experiment results were confirmed in 5 times with 5 different random seeds. |
| Randomization | The training/validation/testing data for downstream tasks were randomly splitted in ratio of 55%:15%:30%. For each patient, we only included the left eye data from one visit to avoid potential bias by inconsistent individual visits. |
| Blinding | When assigning patients randomly to training, validation and testing groups investigators were blinded to patient covariates and all features in the dataset not required to perform the research. |

# Reporting for specific materials, systems and methods

We require information from authors about some types of materials, experimental systems and methods used in many studies. Here, indicate whether each material, system or method listed is relevant to your study. If you are not sure if a list item applies to your research, read the appropriate section before selecting a response.

### Materials & experimental systems

| n/a | Involved in the study |
|---|---|
| ☒ | ☐ Antibodies |
| ☒ | ☐ Eukaryotic cell lines |
| ☒ | ☐ Palaeontology and archaeology |
| ☒ | ☐ Animals and other organisms |
| ☒ | ☐ Clinical data |
| ☒ | ☐ Dual use research of concern |

### Methods

| n/a | Involved in the study |
|---|---|
| ☒ | ☐ ChIP-seq |
| ☒ | ☐ Flow cytometry |
| ☒ | ☐ MRI-based neuroimaging |

