## [Peer Review File · Nature]

Manuscript Title: A foundation model for generalisable disease detection from retinal images

Reviewer Comments & Author Rebuttals

Reviewer Reports on the Initial Version:

Referees' comments:

Referee #1 (Remarks to the Author):

This manuscript is important, well-written, and will be of interest to a wide audience. Methods and data quality are appropriate. Conclusions in line with results. Kudos to the authors for a straightforward, well-executed, and significant study. I mention three points on which authors might choose to expand.

1. Implications of foundation models for medicine

Authors might consider adding a sentence or two expanding on the implications of foundation models for medicine. The term (while not the idea) is relatively recent and some readers might benefit from an expanded explanation of the concept's implications. In particular, it's not just that foundation models can substantially reduce resources required to build usable health care AI (to democratize it, as authors write). It's that in so doing these models could help direct support away from many single use, costly, initially impressive, AI tech demos, that rarely become integrated into routine care and feed skepticism of health care AI's benefits.

2. Potential weaknesses associated with study's data sets

Authors acknowledge limitations associated with relying on datasets drawn almost entirely from the UK. However, because historically this literature (broadly, health care AI) has been prone to exaggeration—enabled often by ignoring limits created by homogeneity of people represented in study datasets—authors might consider highlighting this limitation not only at the end of the paper but also in one of the many passages noting the RETFound's generalizability.

3. What is a clinical task

Claims about the utility of RETFound to perform "a broad range of clinical tasks" (line 172) are accurate in their immediate context. But authors might examine what they mean by clinical tasks. When most people read the phrase, they assume a clinician—a person—is doing something, a task. The clinical tasks to which authors refer in this manuscript are tasks that can be achieved by AI. The point here is not lament AI's potential replacement of people, it is rather to point out that AI is capable at this stage- even informed by foundation models - of performing only a very limited set of clinical tasks. To perform a "broad range" of "diverse" clinical tasks, health care AI needs still to better integrate human-AI collaboration. Authors might consider revising phrases about clinical tasks to more accurately express AI's limitations in this regard.

Minor:

Standardize Foundation model capitalization

Referee #2 (Remarks to the Author):

Healthcare AI is fundamentally constrained by the substantial resources and ethical dilemmas in obtaining sufficiently rich and well characterized training data, compared to the relative overabundance in other fields. Thus for progress in this field, it is imperative to use prior knowledge where available (and reliable) so that the sparse training data can be used most effectively, and both biomarkers based priors - where known - and emergent priors - such as

transfer learning - are gaining interest.

However, the present study does not contribute in a meaningful way to this research endeavor. While it is of interest that the authors show that their transfer approach allows less training data to be used for meaningful accuracy, this is to be expected as transfer learning in all its varieties has been widely published.

ALternatively, the authors could have focused on true validation and transfer learning based algorithms, but their validations are based single physician or other low fidelity reference standards.

Referee #3 (Remarks to the Author):

Foundation models are having a well deserved day in the sun. The promise of general purpose models that can be fine-tuned for specific tasks or subdomains is exciting and the promise seems to be fulfilled with the recent large language models. Therefore testing a foundation model for eye disease like RetFound seems quite appropriate. This is a well written paper which provides suitable motivation and intuitive understanding of the methods for non-ML scientists.

What is clear is that while having at least comparable performance to other pipelines, RetFound is much more label efficient which itself is an important property for the generalization and growth of this model.

In the eye specific disease tasks, the manuscript might seem to be making much larger claims of superior whereas looking at Figure 2b there does not seem to be a single performance that is statistically significant. 2b is important because external validation is the far better measure of robust performance.

3b does have have a couple of significantly superior performances with RetFound. Unless these figures are erroneous this suggests that there is a strong repeated trend towards better performance but it's far from globally significant.

There is quite a lot of diversity in the populations studied and it would be interesting to see an added side-by-side comparison for at least the larger subpopulations to see if the trends seen overall are different for the subgroups. It might be that the self-supervised training might have a better edge there. Or not.

Overall, this manuscript will encourage others to explore the use of Foundation Models in different biomedical domains and with far less dependence on human labels.

Author Rebuttals to Initial Comments: Review responses

Dear editors and reviewers,

Thank you for your encouraging comments and appreciation of research on foundation models for medicine. We have revised the paper in light of your instructive suggestions for a more robust description and insightful analysis (changes highlighted in blue). Please see our point-to-point response below.

Referee #1

This manuscript is important, well-written, and will be of interest to a wide audience. Methods and data quality are appropriate. Conclusions in line with results. Kudos to the authors for a straightforward, well-executed, and significant study. I mention three points on which authors might choose to expand.

1. Implications of foundation models for medicine

Authors might consider adding a sentence or two expanding on the implications of foundation models for medicine. The term (while not the idea) is relatively recent and some readers might benefit from an expanded explanation of the concept's implications. In particular, it's not just that foundation models can substantially reduce resources required to build usable health care AI (to democratize it, as authors write). It's that in so doing these models could help direct support away from many single use, costly, initially impressive, AI tech demos, that rarely become integrated into routine care and feed skepticism of health care AI's benefits.

Response: Thank you for the suggestion. We agree that there is value in expanding on the implications of foundation models for medicine. In particular, your point on the benefits of diverting attention away from superficially impressive but poorly generalisable tech demos is highly relevant to the field of healthcare AI. This has been added to line 257-261 in the discussion.

*By contrast, the data and computational requirements required to fine-tune RETFound to downstream clinical tasks are comparatively small, and therefore more achievable for the majority of institutions. We required only one NVIDIA Tesla T4 (16GB) GPU, requiring about 1.2 hours (\$2) with a dataset of 1000 images. **Moreover, foundational models offer the potential to raise the general quality of healthcare AI models. Their adoption may help avoid superficially impressive models that rarely impact clinical care. These poorly generalisable models consume significant resources and can feed scepticism about the benefits of AI in healthcare.** By making RETFound publicly available, we hope to accelerate the progress of AI in medicine by enabling researchers to utilise our large dataset to design high quality models for use in their own institutions or to explore alternative downstream applications.*

2. Potential weaknesses associated with study's data sets

Authors acknowledge limitations associated with relying on datasets drawn almost entirely from the UK. However, because historically this literature (broadly, health care AI) has been prone to exaggeration—enabled often by ignoring limits created by homogeneity of people represented in study datasets—authors might consider highlighting this limitation not only at the end of the paper but also in one of the many passages noting the RETFound's generalizability.

Response: Thank you for this suggestion. We acknowledge the importance of being transparent about the limitations of our work throughout the manuscript. We have therefore edited our Results section which now also includes this information in line 68-71.

*For the tasks of ocular disease prognosis and systemic disease prediction, we used a cohort from the Moorfields AlzEye study (MEH-AlzEye) which links ophthalmic data of 353,157 patients, who attended Moorfields Eye Hospital between 2008 and 2018, with systemic disease data from hospital admissions across the whole of England. We also used UK Biobank for external evaluation in predicting systemic diseases. **The validation datasets used for ocular disease detection are sourced from multiple countries, while systemic disease prediction was solely validated on UK datasets due to limited availability of this type of longitudinal data. Our assessment of generalisability for systemic disease prediction was therefore based on multiple tasks and datasets, but did not extend to vastly different geographical settings.** Details of the clinical datasets are listed in Supplementary Table 2 (flowcharts for patient selection are listed in Supplementary Fig. 2 to Fig. 6).*

3. What is a clinical task

Claims about the utility of RETFound to perform “a broad range of clinical tasks” (line 172) are accurate in their immediate context. But authors might examine what they mean by clinical tasks. When most people read the phrase, they assume a clinician—a person—is doing something, a task. The clinical tasks to which authors refer in this manuscript are tasks that can be achieved by AI. The point here is not lament AI's potential replacement of people, it is rather to point out that AI is capable at this stage- even informed by foundation models – of performing only a very limited set of clinical tasks. To perform a “broad range” of “diverse” clinical tasks, health care AI needs still to better integrate human-AI collaboration. Authors might consider revising phrases about clinical tasks to more accurately express AI's limitations in this regard.

Response: Thank you for the suggestion. We appreciate that the term ‘clinical task’ may be interpreted in a way that we had not intended. We had merely hoped to convey that the task is something that might be useful in a medical context. We have changed all such occurrences to more neutral phrasing such as ‘disease detection tasks’, which is also consistent with our manuscript title. Additionally, we have included the important point you raised about the need for enhanced human-AI integration to facilitate true diversity for AI applications, in line 277-279.

*Combining these, we propose to further enhance the strength of RETFound in subsequent iterations by introducing even larger quantities of images, exploring additional modalities, enabling dynamic interaction across multi-modal data, and evaluating additional SSL approaches. **Whilst we are optimistic about the broad scope of RETFound to be used for a range of AI tasks, we also acknowledge that enhanced human-AI integration is critical to achieving true diversity in healthcare AI applications.***

Minor:

Standardize Foundation model capitalization

Response: Apologies for this oversight. We have changed all occurrences to 'RETFound' and used the term 'foundation model' (without capitalization) throughout.

Referee #2

Healthcare AI is fundamentally constrained by the substantial resources and ethical dilemmas in obtaining sufficiently rich and well characterized training data, compared to the relative overabundance in other fields. Thus for progress in this field, it is imperative to use prior knowledge where available (and reliable) so that the sparse training data can be used most effectively, and both biomarkers based priors - where known - and emergent priors - such as transfer learning - are gaining interest.

However, the present study does not contribute in a meaningful way to this research endeavor. While it is of interest that the authors show that their transfer approach allows less training data to be used for meaningful accuracy, this is to be expected as transfer learning in all its varieties has been widely published.

Response: Thank you for your comment. After careful consideration, we remain confident that our work offers a significant contribution to the field of healthcare AI. We have carefully reviewed our manuscript and attempted to articulate this contribution more clearly. We hope that these edits and the below discussion will suitably address your concerns. If there are specific aspects of our contribution that you feel are unsubstantiated or poorly expressed, we would be happy to clarify further.

Generally, transfer learning indicates the process of storing knowledge gained whilst solving one machine learning problem and applying it to a different but related problem [1]. In our paper, we refer to the initial stage of transfer learning as 'pre-training' and the subsequent stage as 'fine-tuning.' Traditional supervised-learning-based transfer learning pre-trains AI models with labelled data while recent self-supervised learning requires only unlabelled data, which better fits in healthcare AI where vast amounts of medical data are unlabelled and unexploited. Self-supervised learning with 1.6 million retinal images allows RETFound to learn retina context, including anatomical structures and lesions, which are potential imaging biomarkers for ocular and systemic disease detection (line 143-157 and Figure 5).

Our key contribution is one of the first medical foundation models, which involves the application and validation of self-supervised learning on a large scale. The key advantages of foundation models in medicine are:

1. Reduced requirements for labelled medical data by instead harnessing the vast quantities of unlabelled data that have not previously contributed to training - as you have noted, this is a key barrier to developing medical AI models due to the cost of expert annotations.
2. Improved generalisability to a range of downstream tasks - this is also highly relevant in medicine where AI models may be used for multiple tasks and in multiple settings.
3. Raised standard for medical AI applications allowing the field to move away from costly, single-use, AI 'demos' that are poorly generalisable and feed scepticism over the benefits of AI in healthcare. This final point has been pointed out by Reviewer 1 and we have now included this in our manuscript.

Applying and validating self-supervised learning to a medical setting is an important contribution due to the significant challenges involved. These barriers are outlined in the manuscript and summarised below:

1. Self-supervised learning requires large quantities of domain-specific medical data, often only available at large institutions.
2. Sophisticated data pipelines are required for the data to be curated into a form that is usable, which is expensive to initiate.
3. Training requires enormous computational resources that are not widely available.
4. Diverse downstream tasks should be organised to test the generalisability and efficiency of the foundation model.

By making our model open source, we hope to democratise the development of ophthalmic AI models and accelerate progress towards safe and effective applications. As one of the earliest medical foundation models validated on large-scale clinical data, we are hopeful that ours can be used as an exemplar case to encourage the development of further foundation models within other image-centric specialties, such as dermatology, radiology, and pathology.

Alternatively, the authors could have focused on true validation and transfer learning based algorithms, but their validations are based single physician or other low fidelity reference standards.

Response: Thank you for your comment. Whilst we agree that considering the reliability of reference standards is important, we maintain that those used in this work are well-suited to our study aims. We acknowledge that enhanced clarity would enable readers to evaluate this important aspect more readily and have made appropriate edits. We hope that the below clarifications and manuscript edits will suitably address your concern.

For ocular disease detection, we chose validation sets that are widely-used benchmarks. Rather than single physician grading, 5 out of the 8 validation sets report rigorous adjudication protocols involving multiple clinical experts. The remaining 3 provided limited details such that it is difficult to assess the strength of the labels. These details are summarised below and have now been added to line 360-368 in the Online method.

For systemic disease prediction, validation was performed on 2 large cohort studies linking retinal imaging to incident systemic labels using hospital admissions codes, including the landmark UK Biobank (from which over 6000 papers have been published so far) and the more recent ocular-systemic health cohort, AlzEye. Whilst recognising that the context of collection may impact upon the quality of labelling, numerous epidemiological studies have recognised the benefits of using hospital admissions data as a more pragmatic approach to assessing longitudinal outcomes [2-5], and many studies have found these labels to be sufficiently robust for this type of work [6-8].

Finally, we believe that it is important to consider the context of the work when assessing reference standards, in particular, where the work lies on the translational continuum towards clinical implementation. We see our contribution as fitting early on in this continuum. We seek to provide a resource which enables the democratisation of healthcare AI development, not to provide an autonomous AI solution that is ready to replace clinicians in a real-world setting. We agree that the latter would require a prospective, real-world study,

with rigorous reference standards to guarantee the precision of diagnostic accuracy results. By contrast, the aim of our work is to evaluate RETFound's label efficiency and generalisability compared to existing pre-training approaches. In this context, we believe that permitting the inclusion of datasets with imperfect reference standards is justified when it enables us to better evaluate the generalisability of RETFound to a diverse range of complex tasks and patient populations. It is these superior features of foundation models that promote the potential of applying healthcare AI and human-AI collaboration into routine care.

Summary of grading protocols used for publicly available datasets:	
Dataset	Reference Standard
IDRiD (DR)	2 medical experts provided adjudicated consensus grades
MESSIDOR-2 (DR)	Adjudicated by a panel of 3 retina specialists in accordance with a published protocol
APTOS-2019 (DR)	Kaggle dataset with limited information but possibly a single clinician grader
PAPILA (Glaucoma)	Labelling and segmentation by 2 experts following extensive clinical examination and testing procedure, including retrospective clinical record review
Glaucoma Fundus (Glaucoma)	Agreement of 2 specialists based on visual fields and extensive imaging
JSIEC (Multi-condition)	Labelled by ophthalmologists and confirmed by senior retina specialists. Disagreements resolved by panel of 5 senior retina specialists
Retina (Multi-condition)	Details not available
OCTID (Multi-condition)	Describes image labelling based on the diagnosis of retinal clinical experts but does not specify duplicate adjudication

References

- [1] Pan, Sinno Jialin, and Qiang Yang. "A survey on transfer learning." *IEEE Transactions on knowledge and data engineering* 22.10 (2010): 1345-1359.
- [2] Poplin, R., Varadarajan, A. V., Blumer, K., Liu, Y., McConnell, M. V., Corrado, G. S., ... & Webster, D. R. (2018). Prediction of cardiovascular risk factors from retinal fundus photographs via deep learning. *Nature biomedical engineering*, 2(3), 158-164.
- [3] Diaz-Pinto, A., Ravikumar, N., Attar, R., Suinesiaputra, A., Zhao, Y., Levelt, E., ... & Frangi, A. F. (2022). Predicting myocardial infarction through retinal scans and minimal personal information. *Nature Machine Intelligence*, 4(1), 55-61.
- [4] Han, X., Ong, J. S., An, J., Craig, J. E., Gharahkhani, P., Hewitt, A. W., & MacGregor, S. (2020). Association of myopia and intraocular pressure with retinal detachment in european

descent participants of the UK biobank cohort: a mendelian randomization study. *JAMA ophthalmology*, 138(6), 671-678.

[5] Rudnicka, A. R., Welikala, R., Barman, S., Foster, P. J., Luben, R., Hayat, S., ... & Owen, C. G. (2022). Artificial intelligence-enabled retinal vasculometry for prediction of circulatory mortality, myocardial infarction and stroke. *British Journal of Ophthalmology*, 106(12), 1722-1729.

[6] Wood, A., Denholm, R., Hollings, S., Cooper, J., Ip, S., Walker, V., ... & Sudlow, C. (2021). Linked electronic health records for research on a nationwide cohort of more than 54 million people in England: data resource. *bmj*, 373.

[7] Rannikmäe, K., Ngoh, K., Bush, K., Salman, R. A. S., Doubal, F., Flaig, R., ... & Sudlow, C. L. (2020). Accuracy of identifying incident stroke cases from linked health care data in UK Biobank. *Neurology*, 95(6), e697-e707.

[8] Asaria, P., Elliott, P., Douglass, M., Obermeyer, Z., Soljak, M., Majeed, A., & Ezzati, M. (2017). Acute myocardial infarction hospital admissions and deaths in England: a national follow-back and follow-forward record-linkage study. *The Lancet Public Health*, 2(4), e191-e201.

Reviewer #3

Foundation models are having a well deserved day in the sun. The promise of general purpose models that can be fine-tuned for specific tasks or subdomains is exciting and the promise seems to be fulfilled with the recent large language models. Therefore testing a foundation model for eye disease like RETFound seems quite appropriate. This is a well written paper which provides suitable motivation and intuitive understanding of the methods for non-ML scientists. What is clear is that while having at least comparable performance to other pipelines, RETFound is much more label efficient which itself is an important property for the generalization and growth of this model.

In the eye specific disease tasks, the manuscript might seem to be making much larger claims of superior whereas looking at Figure 2b there does not seem to be a single performance that is statistically significant. 2b is important because external validation is the far better measure of robust performance.

3b does have a couple of significantly superior performances with RETFound. Unless these figures are erroneous this suggests that there is a strong repeated trend towards better performance but it's far from globally significant.

Response: Thank you for your comment. We believe that there has been a miscommunication with regards to the significance results and we apologise for the lack of clarity on our part. We would like to clarify that the results in Figure 2b indicate that RETFound's performance was statistically significantly higher than the most competitive method in *all* cases. As introduced in the Figure legend "Unless otherwise specified, p-value is less than 0.001. ** indicates $0.001 < p\text{-value} < 0.01$, * $0.01 < p\text{-value} < 0.05$, and ns $0.05 < p\text{-value}$." RETFound also significantly outperformed the most competitive method for most of the tasks shown in Figure 3b.

We had initially designed our figures in this way to minimise clutter, but acknowledge that this may not be the most intuitive approach. To avoid any confusion brought by asterisk rules, we have recategorised p-values and changed the way we illustrate significance results as follows:

- ** indicates $p\text{-value} < 0.01$
- * indicates $p\text{-value}$ between $0.01 - 0.05$
- absence of symbol indicates $p > 0.05$ (non-significant)

We have updated all figures with the new asterisk convention. We will continue to work with Nature editors to ensure that we are illustrating these points as effectively as possible and that our figures are consistent with journal guidelines.

Updated Figure 2 and Figure 3 are listed below:

Fig. 2. Performance on ocular disease diagnostic classification. a, internal evaluation, models are adapted to each dataset via fine-tuning and internally evaluated on hold-out test data. The task of diabetic retinopathy is to classify the retinal images based on the International Clinical Diabetic Retinopathy Severity scale, indicating five stages from no diabetic retinopathy to proliferative diabetic retinopathy. The task of glaucoma is to identify images with non-glaucoma, early glaucoma, or advanced glaucoma. The multi-category datasets include multiple retinal diseases (more details are listed in Supplementary Table 1). b, external evaluation, models are fine-tuned on one diabetic retinopathy dataset and externally evaluated on the others. c, performance on ocular disease prognosis. The models are fine-tuned to predict the conversion of fellow eye to wet-AMD in 1 year and evaluated internally. RETFound performs best in all tasks. 95% confidence intervals are shown as error bars. We compare the performance of RETFound with the most competitive comparison model (e.g., SL-ImageNet on APTOS-2019) to check if statistically significant differences exist. p-value is calculated with the two-sided t-test. ** indicates $p\text{-value} < 0.01$ and * indicates $0.01 < p\text{-value} < 0.05$

Fig. 3. Performance on 3-year incidence prediction of systemic diseases with retinal images. a, internal evaluation, models are adapted to curated datasets from MEH-AlzEye via fine-tuning and internally evaluated on hold-out test data. b, external evaluation, models are fine-tuned on MEH-AlzEye and externally evaluated on UK Biobank. Data for internal and external evaluation is described in Supplementary Table 2. Although the overall performances are not high due to the difficulty of tasks, RETFound achieved significantly higher AUROC in all internal evaluation and most external evaluation. We compare the performance of RETFound with the most competitive comparison model to check if statistically significant differences exist. 95% confidence intervals are shown as error bars. ** indicates $p\text{-value} < 0.01$ and * indicates $0.01 < p\text{-value} < 0.05$.

There is quite a lot of diversity in the populations studied and it would be interesting to see an added side-by-side comparison for at least the larger subpopulations to see if the trends seen overall are different for the subgroups. It might be that the self-supervised training might have a better edge there. Or not.

Response: Thank you for this suggestion. We have performed additional subgroup analyses to explore the consistency of our findings within ethnic subpopulations. We selected three systemic disease prediction tasks from the AlzEye dataset for which there were sufficient numbers to permit further analysis, specifically heart failure, ischaemic stroke, and myocardial infarction. Subgroup analyses were performed for White, Asian or Asian British, and Black or Black British subgroups, the three largest major categories of ethnicity as described by the UK Government’s Office for National Statistics.

In each graph, the first column shows the performance on all test data, followed by subgroup results for each of the three ethnic categories. The cohort quantity is listed in the titles. We can see that RETFound significantly outperformed the other pre-training strategies in all cases, and that subgroups showed a similar performance trend compared to the overall cohort.

Incident prediction of heart failure:

Supplementary Fig. 15. AUROC of predicting 3-year heart failure in subsets with different ethnicity, including White, Asian or Asian British, and Black or Black British subgroups, the three largest major categories of ethnicity as described by the UK Government’s Office for National Statistics. Data is from MEH-AlzEye test set. the first column shows the performance on all test data, followed by results on the three subgroups. The cohort quantity is listed in titles. 95% confidence intervals are shown as error bars. We compare the performance of RETFound with the most competitive comparison model to check if statistically significant difference exists. ** indicates $p\text{-value} < 0.01$ and * indicates $0.01 < p\text{-value} < 0.05$.

Incident prediction of ischaemic stroke:

Supplementary Fig. 16. AUROC of predicting 3-year myocardial infarction in subsets with different ethnicity. Data is from MEH-AlzEye test set. the first column shows the performance on all test data, followed by results on White, Asian or Asian British, and Black or Black British cohorts. The cohort quantity is listed in titles. 95% confidence intervals are shown as error bars. We compare the performance of RETFound with the most competitive comparison model to check if statistically significant difference exists. ** indicates $p\text{-value} < 0.01$ and * indicates $0.01 < p\text{-value} < 0.05$.

Incident prediction of myocardial infarction:

Supplementary Fig. 17. AUROC of predicting 3-year ischaemic stroke in subsets with different ethnicity. Data is from MEH-AlzEye test set. the first column shows the performance on all test data, followed by results on White, Asian or Asian British, and Black or Black British cohorts. The cohort quantity is listed in titles. 95% confidence intervals are shown as error bars. We compare the performance of RETFound with the most competitive comparison model to check if statistically significant difference exists. ** indicates $p\text{-value} < 0.01$ and * indicates $0.01 < p\text{-value} < 0.05$.

Overall, this manuscript will encourage others to explore the use of Foundation Models in different biomedical domains and with far less dependence on human labels.

Reviewer Reports on the First Revision:

Referees' comments:

Referee #1 (Remarks to the Author):

Authors' revisions have fully addressed my comments.

Referee #4 (Remarks to the Author):

Zhou et al. present an innovative approach in the field of disease detection from retinal images with their foundation model, RETFound. The authors train RETFound on a substantial dataset of 1.6 million unlabelled retinal images using self-supervised learning (SSL), which allows the model to learn meaningful representations without explicit labels. They subsequently adapt RETFound to specific disease detection tasks using labeled data. The study demonstrates that adapted RETFound consistently outperforms several comparison models in the diagnosis and prognosis of sight-threatening eye diseases. Moreover, the model exhibits promising results in predicting the incidence of complex systemic disorders, such as heart failure and myocardial infarction, while requiring fewer labeled data.

Strengths: The paper's strength lies in its comprehensive exploration of disease detection applications, including ocular disease diagnosis, ocular disease prognosis, and systemic disease prediction. By addressing diverse areas of healthcare, the authors emphasize the potential impact of RETFound in improving medical diagnostics and prognostics. Furthermore, the study pays particular attention to label efficiency, a crucial aspect in the medical domain where acquiring labeled data can be time-consuming and costly. RETFound's ability to achieve impressive performance with limited labeled data signifies its practical utility.

Major Weakness: However, it is worth noting that the idea of leveraging SSL for improved downstream performance in disease diagnosis and related tasks has been explored in previous research. For instance, Truong et al. (reference [1]) demonstrated the effectiveness of self-supervised pre-training models in diabetic retinopathy classification, outperforming supervised ImageNet pre-training models while also considering label efficiency. Additionally, Azizi et al. (reference [2]) investigated the combination of ImageNet pre-training and self-supervised pre-training for tasks such as diabetic macular edema detection, highlighting the benefits in out-of-distribution performance and label efficiency. These existing works raise the question of how RETFound compares to popular contrastive learning approaches commonly used in SSL.

Action points: To strengthen the paper's novelty and appeal to the readership of Nature, I think it is imperative to include a comparative analysis between generative pre-training methods, as employed in this study, and contrastive learning approaches. My suggestion is to include comparisons to 4-5 contrastive learning approaches (perhaps using [1] and [2] to select methods). By addressing this comparison, the authors could determine which method, whether contrastive or generative, offers the most superior universal performance. This addition would enhance the contribution of the paper by providing insights into the relative strengths and limitations of different SSL strategies and facilitating the development of more effective and generalizable disease detection models.

References:

[1] Truong, Tuan, Sadegh Mohammadi, and Matthias Lenga. "How transferable are self-supervised features in medical image classification tasks?." In Machine Learning for Health, pp. 54-74. PMLR, 2021.

[2] Azizi, Shekoofeh, et al. "Robust and efficient medical imaging with self-supervision." arXiv

preprint arXiv:2205.09723 (2022). (a more recent version of this in Nature Biomedical Engineering)

Author Rebuttals to First Revision: Review responses

Dear editors and reviewers,

We would like to express our sincere appreciation for your valuable comments and suggestions. We have carefully revised the manuscript to incorporate your constructive feedback. In particular, we have performed additional analyses using contrastive learning approaches which have allowed us to share new insights into the performance of different SSL strategies (changes are indicated in blue). Please find our point-to-point response below.

Referee #4

Zhou et al. present an innovative approach in the field of disease detection from retinal images with their foundation model, RETFound. The authors train RETFound on a substantial dataset of 1.6 million unlabelled retinal images using self-supervised learning (SSL), which allows the model to learn meaningful representations without explicit labels. They subsequently adapt RETFound to specific disease detection tasks using labeled data. The study demonstrates that adapted RETFound consistently outperforms several comparison models in the diagnosis and prognosis of sight-threatening eye diseases. Moreover, the model exhibits promising results in predicting the incidence of complex systemic disorders, such as heart failure and myocardial infarction, while requiring fewer labeled data.

Strengths: The paper's strength lies in its comprehensive exploration of disease detection applications, including ocular disease diagnosis, ocular disease prognosis, and systemic disease prediction. By addressing diverse areas of healthcare, the authors emphasize the potential impact of RETFound in improving medical diagnostics and prognostics. Furthermore, the study pays particular attention to label efficiency, a crucial aspect in the medical domain where acquiring labeled data can be time-consuming and costly. RETFound's ability to achieve impressive performance with limited labeled data signifies its practical utility.

Response: Thank you for your recognition of RETFound's promising performance and label efficiency, as well as its practical utility across diverse disease detection applications.

Major Weakness: However, it is worth noting that the idea of leveraging SSL for improved downstream performance in disease diagnosis and related tasks has been explored in previous research. For instance, Truong et al. (reference [1]) demonstrated the effectiveness of self-supervised pre-training models in diabetic retinopathy classification, outperforming supervised ImageNet pre-training models while also considering label efficiency. Additionally, Azizi et al. (reference [2]) investigated the combination of ImageNet pre-training and self-supervised pre-training for tasks such as diabetic macular edema detection, highlighting the benefits in out-of-distribution performance and label efficiency. These

existing works raise the question of how RETFound compares to popular contrastive learning approaches commonly used in SSL.

Response: Thank you for your valuable comments and recommendation of the work of Truong et al. [30] and Azizi et al. [26]. We believe they each represent important contributions to this rapidly evolving field. While recognising the strengths of these papers, we believe that RETFound builds upon them in several important ways.

Truong et al. applied SSL techniques to various medical classification tasks, including diabetic retinopathy, and demonstrated that SSL pre-trained models yield richer embeddings compared to supervised models. Notably, their pre-training data was limited to natural images from ImageNet. RETFound makes further progress by applying SSL techniques to a large unlabelled medical dataset of 1.6 million images (in addition to ImageNet), showing that the inclusion of medical images improves downstream performance. This is a notable contribution since it requires access to a large clinical dataset and a well-developed pipeline for incorporating images into a model.

Azizi et al. applied SSL techniques to unlabelled medical data and show improved performance compared to a supervised approach. A key strength of this paper is that their approach is demonstrated across five imaging types that are commonly used in medicine. However, their downstream evaluation is limited to only one or two tasks for each imaging type.

The major advance we report with RETFound is the ability to support 13 wide-ranging downstream evaluation tasks from a single model using two common retinal imaging modalities: fundus photography and optical coherence tomography (OCT). These tasks cover ophthalmic disease detection and prognosis, as well as systemic (non-ophthalmic) disease prediction. Thus RETFound demonstrates the key potential of foundation models in medical imaging for the first time - the ability to learn generalisable features applicable to diverse downstream tasks. Demonstrating this benefit on a diverse range of tasks relies on our uniquely extensive data linkage between retinal imaging and longitudinal ocular and systemic health outcomes.

We will make RETFound openly available to accelerate research in this domain, enabling RETFound to serve as the pioneering foundation model in healthcare and supporting the creation of comprehensive foundation models in other medical specialties.

We have made revisions to our manuscript to acknowledge these prior contributions and highlight the additional advances brought about by RETFound (line 29-37):

“Although several studies have shown that SSL can increase performance for individual ocular disease detection tasks, such as diabetic macular edema²⁶, age-related macular degeneration (AMD)²⁷, and referable diabetic retinopathy^{28,29,30}, there has been limited work demonstrating the ability of a single SSL pre-trained model to generalise to a diverse range of complex tasks. Progress has likely been hampered by the challenges involved with curating a large repository of retinal images with extensive linkage to multiple relevant disease outcomes. Moreover, the capabilities of different SSL approaches (contrastive SSL versus generative SSL) and the

interpretability of SSL models in retinal imaging, remain relatively under-explored. Developing an understanding of the specific features that SSL models learn during training is an important step for safe and reliable translation to clinical practice.”

While we remain confident in RETFound’s widespread potential for impact, we also acknowledge that conducting additional experiments to compare contrastive and generative SSL approaches would further enhance the scope of our work, as described in the following section.

Action points: To strengthen the paper's novelty and appeal to the readership of Nature, I think it is imperative to include a comparative analysis between generative pre-training methods, as employed in this study, and contrastive learning approaches. My suggestion is to include comparisons to 4-5 contrastive learning approaches (perhaps using [1] and [2] to select methods). By addressing this comparison, the authors could determine which method, whether contrastive or generative, offers the most superior universal performance. This addition would enhance the contribution of the paper by providing insights into the relative strengths and limitations of different SSL strategies and facilitating the development of more effective and generalizable disease detection models.

Response: Thank you for the suggestion. We fully agree that comparing and discussing how different SSL strategies (e.g. contrastive SSL versus generative SSL) perform within our RETFound training and evaluation pipeline will enhance the paper and increase its impact.

Following your recommendations, we have now included four contrastive learning techniques for comparison: SimCLR¹⁶, SwAV³⁶, DINO³⁷, and MoCo-v3¹⁴. We substituted the generative SSL method (i.e. masked autoencoder) with each contrastive SSL approach in the RETFound framework. We have added related content to the paper, including

- a) *some context for the new experiment;*
“Moreover, the capabilities of different SSL approaches (contrastive vs generative) and the interpretability of SSL models in retinal imaging, remain relatively under-explored.” (line 34-35)

“Additionally, we explored the performance of using different SSL strategies, i.e. generative SSL versus contrastive SSL approaches, by substituting the primary SSL technique (i.e. masked autoencoder) with SimCLR¹⁶, SwAV³⁶, DINO³⁷, and MoCo-v3¹⁴ within the RETFound framework.” (line 85-87)
- b) *new quantitative results in Supplementary Table 4 and bar plots in Supplementary Fig.15 (Supplementary Table 3 is also attached for reference);*
- c) *a subsection in the Results;*
“Performance of different SSL strategies in the RETFound framework. We explored the performance of different SSL strategies, i.e. generative SSL (e.g. masked autoencoder) and contrastive SSL (e.g. SimCLR, SwAV, DINO, and MoCo-v3), in the RETFound framework. As shown in Supplementary Table 4, RETFound

with different contrastive SSL strategies showed decent performance in downstream tasks. For instance, RETFound with DINO achieved AUROC of 0.866 (95% CI 0.864, 0.869) and 0.728 (95% CI 0.725, 0.731) respectively on wet-AMD prognosis and ischaemic stroke prediction, outperforming the baseline SL-ImageNet reported in Supplementary Table 3. This demonstrates the effectiveness of RETFound framework with diverse SSL strategies. Among these SSL strategies, the masked autoencoder (RETFound in Supplementary Table 3) performed significantly better than the contrastive learning approaches in most disease detection tasks (Supplementary Fig. 15). All quantitative results are listed in Supplementary Table 3 and 4.” (line 180-189)

d) related discussion;

“We observe that RETFound maintains competitive performance for disease detection tasks, even when substituting various contrastive SSL approaches into the framework. It appears that the generative approach using the masked autoencoder generally outperforms the contrastive approaches, including SwAV, SimCLR, MoCo-v3, and DINO. However, it is worth noting that asserting the superiority of the masked autoencoder requires caution, given the presence of several variables across all models, such as network architectures (e.g. ResNet-50 for SwAV and SimCLR, Transformers for the others) and hyperparameters (e.g. learning rate scheduler). Our comparison demonstrates that the combination of powerful network architecture and complex pretext tasks can produce effective and general-purpose medical foundation models, aligning with the insights derived from large language models in healthcare^{49,50}. Additionally, the comparison further supports the notion that the retinal-specific context learned from the masked autoencoder's pretext task, which includes anatomical structures like the optic nerve head and retinal nerve fibre layer (as shown in Fig. 5a), indeed provides discriminative information for the detection of ocular and systemic diseases.” (line 266-277)

e) implementation in Online Method;

“**Contrastive learning approach implementation.** We substitute the primary SSL approach (i.e. masked autoencoder) with SimCLR¹⁶, SwAV³⁷, DINO³⁸, and MoCo-v3¹⁴ in the RETFound framework to produce variants of the pre-trained model for comparison. For SSL training with each contrastive learning approach, we follow the recommended network architectures and hyperparameter settings from the published papers for optimal performance. We first load the pre-trained weights on ImageNet-1k to the models and further train the models with 1.6 million retinal images with each contrastive learning approach, so as to obtain pre-trained models. We then follow the identical process of transferring the masked autoencoder to fine-tune those pre-trained models for the downstream disease detection tasks.” (line 457-464)

“**Computational resources.** ...We allocate an equal computational cost to each SSL approach for pre-training....” (line 468-469)

References:

[30] Truong, Tuan, Sadegh Mohammadi, and Matthias Lenga. "How transferable are self-supervised features in medical image classification tasks?." In Machine Learning for Health, pp. 54-74. PMLR, 2021.

[26] Azizi, Shekoofeh, et al. "Robust and efficient medical imaging with self-supervision." arXiv preprint arXiv:2205.09723 (2022). (a more recent version of this in Nature Biomedical Engineering)

[16] Chen, Ting, et al. "A simple framework for contrastive learning of visual representations." International conference on machine learning. PMLR, 2020.

[14] Chen, Xinlei, Saining Xie, and Kaiming He. "An empirical study of training self-supervised vision transformers." Proceedings of the IEEE/CVF International Conference on Computer Vision. 2021.

[36] Caron, Mathilde, et al. "Unsupervised learning of visual features by contrasting cluster assignments." Advances in neural information processing systems 33 (2020): 9912-9924.

[37] Caron, Mathilde, et al. "Emerging properties in self-supervised vision transformers." Proceedings of the IEEE/CVF international conference on computer vision. 2021.

[49] Singhal, Karan, et al. "Large Language Models Encode Clinical Knowledge." arXiv preprint arXiv:2212.13138 (2022).

[50] Singhal, Karan, et al. "Towards Expert-Level Medical Question Answering with Large Language Models." arXiv preprint arXiv:2305.09617 (2023).

Supplementary Fig. 15. AUROC of predicting ocular diseases and systemic diseases by the models pre-trained with different SSL strategies, including the masked autoencoder (MAE), SwAV, SimCLR, MoCo-v3, and DINO. The data for systemic disease tasks is from MEH-AlzEye dataset. RETFound with MAE achieved significantly higher AUROC in most tasks. All quantitative results of RETFound using MAE can be found in Supplementary Table 3. The corresponding results for the contrastive SSL approaches are listed in Supplementary Table 4. 95% confidence intervals are shown as error bars. We compare the performance of MAE with the most competitive comparison model to check if statistically significant difference exists. ** indicates $p\text{-value} < 0.01$ and * indicates $0.01 < p\text{-value} < 0.05$.

Supplementary Table 4. Performance of contrastive SSL models fine-tuned to multiple disease detection and prediction. Parentheses include values of 95% confidence interval. For prediction of ischaemic stroke, myocardial infarction, heart failure and Parkinson’s disease, performance on MEH-AlzEye indicates internal evaluation while performance on UK Biobank for external evaluation. The results of RETFound using the masked autoencoder can be found in Supplementary Table 3.

Metric	AUROC				AUPR			
Comparison groups	DINO	MoCo-v3	SimCLR	SwAV	DINO	MoCo-v3	SimCLR	SwAV
3-year incidence prediction of ischaemic stroke								
MEH-AlzEye CFP	0.728 (0.725, 0.731)	0.695 (0.682, 0.708)	0.674 (0.673, 0.674)	0.682 (0.663, 0.701)	0.726 (0.722, 0.73)	0.689 (0.675, 0.703)	0.649 (0.644, 0.655)	0.658 (0.638, 0.677)
UK Biobank CFP	0.625 (0.616, 0.634)	0.557 (0.539, 0.575)	0.542 (0.524, 0.561)	0.549 (0.538, 0.56)	0.629 (0.619, 0.639)	0.56 (0.542, 0.577)	0.535 (0.515, 0.555)	0.533 (0.519, 0.546)
MEH-AlzEye OCT	0.725 (0.711, 0.739)	0.712 (0.665, 0.759)	0.667 (0.639, 0.693)	0.601 (0.583, 0.618)	0.715 (0.701, 0.729)	0.702 (0.649, 0.754)	0.648 (0.629, 0.667)	0.578 (0.56, 0.594)
UK Biobank OCT	0.59 (0.561, 0.618)	0.549 (0.525, 0.572)	0.492 (0.481, 0.504)	0.501 (0.479, 0.521)	0.586 (0.556, 0.617)	0.556 (0.535, 0.577)	0.496 (0.488, 0.504)	0.502 (0.482, 0.523)

3-year incidence prediction of myocardial infarction								
MEH-AlzEye CFP	0.731 (0.727, 0.734)	0.661 (0.645, 0.675)	0.644 (0.639, 0.649)	0.65 (0.636, 0.664)	0.722 (0.719, 0.725)	0.653 (0.637, 0.669)	0.612 (0.599, 0.624)	0.625 (0.614, 0.637)
UK Biobank CFP	0.611 (0.605, 0.616)	0.568 (0.543, 0.594)	0.563 (0.545, 0.58)	0.513 (0.501, 0.526)	0.609 (0.604, 0.614)	0.557 (0.537, 0.577)	0.551 (0.533, 0.568)	0.513 (0.499, 0.527)
MEH-AlzEye OCT	0.708 (0.701, 0.717)	0.7 (0.675, 0.725)	0.658 (0.65, 0.665)	0.598 (0.589, 0.606)	0.693 (0.684, 0.703)	0.687 (0.66, 0.714)	0.644 (0.632, 0.655)	0.573 (0.563, 0.582)
UK Biobank OCT	0.583 (0.578, 0.589)	0.536 (0.494, 0.579)	0.583 (0.557, 0.609)	0.501 (0.489, 0.513)	0.567 (0.562, 0.572)	0.546 (0.507, 0.585)	0.573 (0.552, 0.594)	0.501 (0.492, 0.51)
3-year incidence prediction of heart failure								
MEH-AlzEye CFP	0.785 (0.783, 0.786)	0.745 (0.738, 0.751)	0.714 (0.699, 0.729)	0.72 (0.703, 0.736)	0.773 (0.771, 0.775)	0.731 (0.724, 0.738)	0.692 (0.681, 0.703)	0.691 (0.667, 0.715)
UK Biobank CFP	0.656 (0.648, 0.663)	0.634 (0.621, 0.648)	0.606 (0.582, 0.63)	0.528 (0.501, 0.555)	0.65 (0.644, 0.657)	0.629 (0.614, 0.644)	0.587 (0.569, 0.605)	0.528 (0.509, 0.546)

MEH-AlzEye OCT	0.793 (0.791, 0.795)	0.8 (0.796, 0.803)	0.743 (0.731, 0.755)	0.655 (0.642, 0.667)	0.782 (0.781, 0.783)	0.791 (0.787, 0.795)	0.724 (0.706, 0.742)	0.624 (0.611, 0.637)
UK Biobank OCT	0.686 (0.678, 0.695)	0.672 (0.654, 0.69)	0.601 (0.569, 0.631)	0.509 (0.493, 0.525)	0.677 (0.667, 0.686)	0.661 (0.638, 0.684)	0.588 (0.557, 0.618)	0.508 (0.494, 0.522)
3-year incidence prediction of Parkinson's disease								
MEH-AlzEye CFP	0.636 (0.609, 0.663)	0.638 (0.624, 0.651)	0.631 (0.587, 0.674)	0.59 (0.564, 0.615)	0.657 (0.629, 0.685)	0.644 (0.632, 0.656)	0.626 (0.584, 0.667)	0.565 (0.537, 0.593)
UK Biobank CFP	0.534 (0.476, 0.593)	0.499 (0.465, 0.534)	0.523 (0.483, 0.562)	0.481 (0.389, 0.572)	0.585 (0.539, 0.631)	0.549 (0.512, 0.585)	0.563 (0.546, 0.58)	0.507 (0.441, 0.573)
MEH-AlzEye OCT	0.661 (0.633, 0.688)	0.686 (0.639, 0.733)	0.609 (0.55, 0.667)	0.499 (0.462, 0.536)	0.652 (0.63, 0.673)	0.698 (0.654, 0.743)	0.585 (0.54, 0.629)	0.507 (0.473, 0.541)
UK Biobank OCT	0.447 (0.398, 0.495)	0.585 (0.548, 0.623)	0.498 (0.493, 0.503)	0.441 (0.398, 0.484)	0.488 (0.452, 0.524)	0.613 (0.585, 0.641)	0.5 (0.495, 0.506)	0.47 (0.442, 0.497)
1-year fellow eye converting to wet-AMD								

MEH-AlzEye CFP	0.866 (0.864, 0.869)	0.828 (0.819, 0.837)	0.805 (0.789, 0.822)	0.803 (0.789, 0.817)	0.845 (0.844, 0.847)	0.819 (0.808, 0.83)	0.782 (0.767, 0.796)	0.781 (0.768, 0.793)
MEH-AlzEye OCT	0.775 (0.76, 0.789)	0.763 (0.753, 0.774)	0.664 (0.636, 0.691)	0.638 (0.621, 0.654)	0.773 (0.756, 0.789)	0.76 (0.748, 0.772)	0.644 (0.623, 0.665)	0.611 (0.6, 0.621)
Diabetic retinopathy								
MESSIDOR-2	0.835 (0.831, 0.838)	0.847 (0.84, 0.853)	0.833 (0.816, 0.849)	0.826 (0.804, 0.849)	0.512 (0.499, 0.525)	0.49 (0.473, 0.507)	0.579 (0.549, 0.61)	0.551 (0.521, 0.581)
IDRID	0.756 (0.742, 0.77)	0.763 (0.751, 0.775)	0.762 (0.748, 0.776)	0.76 (0.739, 0.781)	0.464 (0.444, 0.483)	0.456 (0.44, 0.472)	0.451 (0.425, 0.478)	0.459 (0.437, 0.48)
Kaggle APTOS- 2019	0.932 (0.932, 0.932)	0.931 (0.928, 0.934)	0.912 (0.908, 0.916)	0.913 (0.906, 0.921)	0.666 (0.662, 0.671)	0.676 (0.668, 0.683)	0.599 (0.584, 0.614)	0.606 (0.588, 0.624)
Glaucoma								
PAPILA	0.788 (0.772, 0.805)	0.797 (0.76, 0.834)	0.805 (0.773, 0.837)	0.791 (0.774, 0.809)	0.635 (0.62, 0.651)	0.627 (0.568, 0.687)	0.637 (0.591, 0.683)	0.617 (0.589, 0.644)

Glaucoma Fundus	0.918 (0.915, 0.921)	0.915 (0.911, 0.918)	0.897 (0.884, 0.911)	0.919 (0.908, 0.931)	0.802 (0.795, 0.809)	0.839 (0.833, 0.845)	0.765 (0.737, 0.793)	0.798 (0.768, 0.827)
Multi-class disease								
JSIEC	0.958 (0.955, 0.961)	0.954 (0.951, 0.957)	0.956 (0.955, 0.958)	0.951 (0.947, 0.954)	0.652 (0.633, 0.671)	0.755 (0.742, 0.769)	0.822 (0.812, 0.833)	0.758 (0.743, 0.773)
Retina	0.787 (0.778, 0.796)	0.787 (0.767, 0.806)	0.817 (0.81, 0.825)	0.775 (0.764, 0.785)	0.608 (0.591, 0.624)	0.635 (0.604, 0.666)	0.633 (0.618, 0.647)	0.581 (0.56, 0.602)
OCTID	0.961 (0.959, 0.962)	0.936 (0.934, 0.937)	0.96 (0.955, 0.965)	0.948 (0.94, 0.956)	0.937 (0.932, 0.942)	0.837 (0.833, 0.841)	0.927 (0.914, 0.94)	0.906 (0.874, 0.938)

Supplementary Table 3. Performance of models fine-tuned to multiple disease detection and prediction. Parentheses include values of 95% confidence interval. For prediction of ischaemic stroke, myocardial infarction, heart failure and Parkinson’s disease, performance on MEH-AlzEye indicates internal evaluation while performance on UK Biobank for external evaluation. **(No change has been made on Supplementary Table 3. It is attached for reference)**

Metric	AUROC				AUPR			
Comparison groups	RETFound	SSL-Retinal	SSL-ImageNet	SL-ImageNet	RETFound	SSL-Retinal	SSL-ImageNet	SL-ImageNet
3-year incidence prediction of ischaemic stroke								
MEH-AlzEye CFP	0.754 (0.752, 0.756)	0.697 (0.689, 0.706)	0.684 (0.679, 0.689)	0.665 (0.655, 0.674)	0.744 (0.74, 0.749)	0.685 (0.677, 0.693)	0.676 (0.671, 0.681)	0.669 (0.66, 0.678)
UK Biobank CFP	0.594 (0.58, 0.608)	0.581 (0.569, 0.593)	0.57 (0.565, 0.574)	0.547 (0.54, 0.554)	0.587 (0.575, 0.601)	0.571 (0.562, 0.58)	0.564 (0.562, 0.567)	0.543 (0.534, 0.552)
MEH-AlzEye OCT	0.746 (0.742, 0.749)	0.701 (0.699, 0.702)	0.648 (0.632, 0.664)	0.678 (0.669, 0.687)	0.736 (0.731, 0.741)	0.683 (0.68, 0.686)	0.642 (0.622, 0.661)	0.666 (0.657, 0.674)
UK Biobank OCT	0.559 (0.541, 0.577)	0.551 (0.545, 0.558)	0.501 (0.484, 0.517)	0.54 (0.531, 0.549)	0.547 (0.529, 0.566)	0.533 (0.529, 0.536)	0.507 (0.497, 0.517)	0.548 (0.54, 0.556)

3-year incidence prediction of myocardial infarction								
MEH-AlzEye CFP	0.737 (0.731, 0.743)	0.672 (0.67, 0.675)	0.619 (0.61, 0.628)	0.611 (0.605, 0.617)	0.726 (0.72, 0.733)	0.641 (0.638, 0.644)	0.579 (0.567, 0.592)	0.592 (0.586, 0.598)
UK Biobank CFP	0.579 (0.566, 0.593)	0.509 (0.49, 0.528)	0.498 (0.485, 0.511)	0.499 (0.493, 0.504)	0.563 (0.551, 0.575)	0.48 (0.469, 0.491)	0.503 (0.494, 0.512)	0.505 (0.497, 0.514)
MEH-AlzEye OCT	0.731 (0.725, 0.736)	0.682 (0.677, 0.687)	0.615 (0.605, 0.625)	0.633 (0.627, 0.639)	0.726 (0.718, 0.734)	0.67 (0.663, 0.678)	0.602 (0.592, 0.613)	0.631 (0.625, 0.638)
UK Biobank OCT	0.605 (0.59, 0.621)	0.586 (0.574, 0.597)	0.498 (0.486, 0.51)	0.536 (0.531, 0.541)	0.601 (0.59, 0.612)	0.589 (0.576, 0.602)	0.507 (0.499, 0.514)	0.544 (0.538, 0.551)
3-year incidence prediction of heart failure								
MEH-AlzEye CFP	0.794 (0.792, 0.797)	0.715 (0.714, 0.716)	0.715 (0.713, 0.716)	0.705 (0.704, 0.706)	0.784 (0.779, 0.789)	0.699 (0.698, 0.701)	0.708 (0.707, 0.709)	0.692 (0.691, 0.693)
UK Biobank CFP	0.676 (0.67, 0.682)	0.61 (0.603, 0.617)	0.585 (0.581, 0.59)	0.59 (0.583, 0.597)	0.655 (0.647, 0.663)	0.594 (0.587, 0.601)	0.564 (0.561, 0.567)	0.582 (0.576, 0.588)

MEH-AlzEye OCT	0.809 (0.807, 0.81)	0.768 (0.767, 0.769)	0.734 (0.732, 0.736)	0.727 (0.725, 0.73)	0.801 (0.798, 0.804)	0.762 (0.76, 0.763)	0.721 (0.72, 0.723)	0.716 (0.714, 0.719)
UK Biobank OCT	0.682 (0.678, 0.685)	0.647 (0.644, 0.65)	0.612 (0.609, 0.614)	0.602 (0.596, 0.608)	0.684 (0.68, 0.688)	0.644 (0.641, 0.646)	0.606 (0.604, 0.608)	0.587 (0.584, 0.589)
3-year incidence prediction of Parkinson's disease								
MEH-AlzEye CFP	0.669 (0.65, 0.688)	0.591 (0.564, 0.618)	0.595 (0.584, 0.606)	0.597 (0.581, 0.613)	0.662 (0.641, 0.683)	0.606 (0.587, 0.624)	0.606 (0.599, 0.612)	0.586 (0.565, 0.608)
UK Biobank CFP	0.57 (0.554, 0.586)	0.501 (0.466, 0.536)	0.476 (0.458, 0.494)	0.517 (0.491, 0.543)	0.598 (0.561, 0.637)	0.525 (0.488, 0.561)	0.514 (0.488, 0.54)	0.552 (0.526, 0.578)
MEH-AlzEye OCT	0.758 (0.738, 0.777)	0.688 (0.676, 0.699)	0.682 (0.657, 0.707)	0.633 (0.611, 0.656)	0.765 (0.742, 0.789)	0.713 (0.705, 0.721)	0.684 (0.657, 0.71)	0.631 (0.609, 0.652)
UK Biobank OCT	0.551 (0.534, 0.567)	0.503 (0.488, 0.519)	0.522 (0.487, 0.556)	0.508 (0.482, 0.535)	0.531 (0.513, 0.547)	0.493 (0.475, 0.511)	0.501 (0.481, 0.521)	0.513 (0.491, 0.534)
1-year fellow eye converting to wet-AMD								

MEH-AlzEye CFP	0.862 (0.86, 0.865)	0.814 (0.809, 0.819)	0.831 (0.826, 0.836)	0.83 (0.825, 0.836)	0.849 (0.845, 0.853)	0.795 (0.791, 0.8)	0.823 (0.819, 0.827)	0.833 (0.827, 0.839)
MEH-AlzEye OCT	0.799 (0.796, 0.802)	0.783 (0.778, 0.788)	0.773 (0.769, 0.777)	0.756 (0.753, 0.759)	0.789 (0.784, 0.794)	0.783 (0.779, 0.788)	0.766 (0.76, 0.772)	0.746 (0.744, 0.748)
Diabetic retinopathy								
MESSIDOR-2	0.884 (0.88, 0.887)	0.816 (0.814, 0.818)	0.823 (0.809, 0.837)	0.859 (0.854, 0.864)	0.669 (0.656, 0.683)	0.456 (0.451, 0.46)	0.48 (0.445, 0.516)	0.601 (0.591, 0.612)
IDRID	0.822 (0.815, 0.829)	0.703 (0.695, 0.71)	0.76 (0.746, 0.773)	0.778 (0.77, 0.786)	0.496 (0.481, 0.511)	0.402 (0.395, 0.409)	0.449 (0.423, 0.475)	0.499 (0.48, 0.519)
Kaggle APTOS- 2019	0.943 (0.941, 0.944)	0.899 (0.898, 0.9)	0.894 (0.891, 0.896)	0.918 (0.916, 0.919)	0.726 (0.721, 0.73)	0.636 (0.633, 0.639)	0.679 (0.666, 0.692)	0.704 (0.694, 0.715)
Glaucoma								
PAPILA	0.855 (0.842, 0.868)	0.705 (0.685, 0.725)	0.727 (0.709, 0.745)	0.819 (0.807, 0.83)	0.748 (0.706, 0.791)	0.507 (0.475, 0.539)	0.521 (0.488, 0.554)	0.66 (0.641, 0.679)

Glaucoma Fundus	0.943 (0.941, 0.945)	0.895 (0.892, 0.897)	0.885 (0.881, 0.889)	0.919 (0.917, 0.922)	0.863 (0.86, 0.867)	0.809 (0.807, 0.81)	0.792 (0.784, 0.799)	0.813 (0.809, 0.817)
Multi-class disease								
JSIEC	0.99 (0.989, 0.991)	0.929 (0.927, 0.931)	0.949 (0.948, 0.95)	0.975 (0.974, 0.977)	0.884 (0.878, 0.889)	0.605 (0.592, 0.619)	0.816 (0.808, 0.825)	0.89 (0.882, 0.897)
Retina	0.847 (0.841, 0.853)	0.703 (0.673, 0.733)	0.699 (0.69, 0.708)	0.831 (0.824, 0.838)	0.697 (0.691, 0.703)	0.505 (0.483, 0.527)	0.497 (0.478, 0.517)	0.647 (0.638, 0.657)
OCTID	0.998 (0.998, 0.999)	0.964 (0.963, 0.966)	0.961 (0.958, 0.963)	0.965 (0.964, 0.966)	0.993 (0.989, 0.998)	0.956 (0.951, 0.96)	0.942 (0.93, 0.954)	0.962 (0.96, 0.965)

Reviewer Reports on the Second Revision:

Referees' comments:

Referee #4 (Remarks to the Author):

Thank you for your thorough and thoughtful response to my review. I commend your efforts in addressing the comments and incorporating the suggested changes into the manuscript.

The inclusion of the comparative analysis between generative and contrastive learning approaches is a valuable addition to the paper. I am pleased to see that you have included four contrastive learning techniques for comparison and provided quantitative results and discussions on their performance within the RETFound framework. This comparative analysis enhances the novelty and impact of the paper by providing insights into the relative strengths and limitations of different SSL strategies. I would personally want to see at least one of the contrastive models brought up to the main text (Figures 2 and 3), but this is not a requirement.

I also appreciate the revisions made in the manuscript to acknowledge prior contributions and highlight the additional advancements brought about by RETFound.

Overall, I believe that the revisions and additions you have made in response to my review have significantly improved the manuscript. The inclusion of the comparative analysis and the discussions on the different SSL strategies further strengthen the paper's contribution and make it more appealing to the readership. Thank you for addressing my concerns and for your diligence in enhancing the quality of the manuscript.

Author Rebuttals to Second Revision:

Referee #4

Thank you for your thorough and thoughtful response to my review. I commend your efforts in addressing the comments and incorporating the suggested changes into the manuscript.

The inclusion of the comparative analysis between generative and contrastive learning approaches is a valuable addition to the paper. I am pleased to see that you have included four contrastive learning techniques for comparison and provided quantitative results and discussions on their performance within the RETFound framework. This comparative analysis enhances the novelty and impact of the paper by providing insights into the relative strengths and limitations of different SSL strategies. I would personally want to see at least one of the contrastive models brought up to the main text (Figures 2 and 3), but this is not a requirement.

I also appreciate the revisions made in the manuscript to acknowledge prior contributions and highlight the additional advancements brought about by RETFound.

Overall, I believe that the revisions and additions you have made in response to my review have significantly improved the manuscript. The inclusion of the comparative analysis and the discussions on the different SSL strategies further strengthen the paper's contribution and make it more appealing to the readership. Thank you for addressing my concerns and for your diligence in enhancing the quality of the manuscript.

Response: Thank you again for your insightful suggestions. We also appreciate your recognition of our efforts in paper revision and the improvement on the paper. We have moved SSL comparison results to Figure 6 for a wider readership and impact.